# DATA-FREE ASYMPTOTICS-INFORMED OPERATOR NETWORKS FOR SINGULARLY PERTURBED PDES

## ABSTRACT

Recent advancements in machine learning (ML) have shown promise in solving partial differential equations (PDEs), but significant challenges remain, particularly in handling complex scenarios. Singularly perturbed differential equations present unique computational difficulties due to rapid transitions within thin boundary or interior layers, where ML methods often struggle. Moreover, these problems require massive adaptive mesh refinement, making dataset generation computationally expensive. In this paper, we introduce eFEONet, an enriched Finite Element Operator Network designed to overcome these challenges. By leveraging singular perturbation analysis from PDE theory, eFEONet incorporates special basis functions that capture the asymptotic behavior of solutions, enabling accurate modeling of sharp transitions. Our approach is highly data-efficient, requiring minimal training data or even functioning without a dataset. Furthermore, we provide a rigorous convergence analysis and empirically validate eFEONet across various boundary and interior layer problems.

## 1 INTRODUCTION

The use of machine learning (ML) to solve partial differential equations (PDEs) has made significant advancements in recent years, offering innovative approaches to tackle longstanding challenges in scientific computing (Lagaris et al., 1998; Lu et al., 2021b; Yu et al., 2018; Ainsworth & Dong, 2021). Among these methods, operator networks have emerged as a practical and efficient tool due to their ability to infer solutions quickly after training (Lu et al., 2021a; Li et al., 2021a). Unlike classical numerical methods that iteratively solve PDEs for each parameter setting, operator networks learn the solution operator itself, enabling rapid prediction and establishing a new paradigm for parametric PDEs. Their application, however, faces key challenges. Training typically requires precomputed datasets generated by conventional numerical solvers, a process that is computationally costly for complex PDEs. Singularly perturbed equations are particularly difficult, as sharp transitions within thin boundary or interior layers demand expensive, high-fidelity datasets and often degrade operator network performance due to their reliance on smooth priors (Lu et al., 2022).

Boundary and interior layer phenomena are of paramount importance in many scientific and engineering disciplines, including fluid dynamics, biology, and chemical reactions (Schlichting & Gersten,

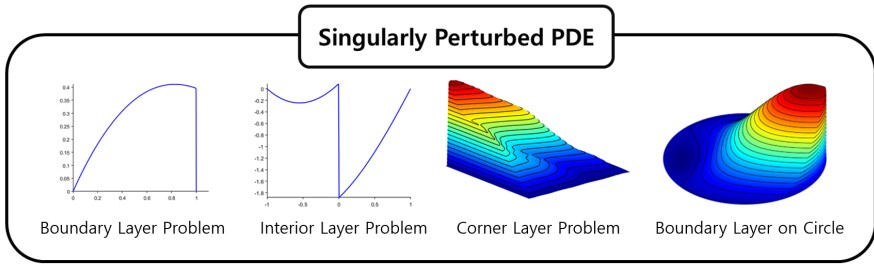

Figure 1: Representative solution profiles for singularly perturbed PDEs, illustrating the inherent stiffness of boundary and interior layers across various domains. The sharp gradients and rapid transitions depicted here highlight the intrinsic stiffness and associated computational challenges.

2016; Batchelor, 2000). These problems are characterized by sharp changes in solution profiles within thin layers, making them notoriously difficult to handle even with advanced numerical methods. The challenge arises from the small diffusive parameter $\varepsilon > 0$ in these equations, which leads to steep gradients over small spatial regions. See Figure 2 where the examples of interior layer phenomena are presented. Developing methods to accurately and efficiently solve such problems remains a challenging task in scientific computing (Zienkiewicz & Taylor, 2000; Hughes, 2000). Machine learning-based approaches face additional challenges because they are inherently better at learning smooth functions but struggle to accurately capture sharp transitions. Neural networks, for instance, are often designed to approximate solutions that vary gradually, making it difficult to capture the steep gradients and sharp transitions characteristic of boundary or interior layers (Karniadakis et al., 2021). These boundary and interior layer problems require computationally expensive massive mesh refinement to obtain accurate solutions. Moreover, as $\varepsilon > 0$ decreases, the mesh size must become finer, following an approximate scaling of *adaptive mesh size* $\simeq \varepsilon$. This results in a significant drawback, as data generation can become prohibitively expensive in many cases. This limitation highlights the need for new architectures and methodologies that can handle these complexities without compromising accuracy or efficiency.

In this paper, we propose an enriched Finite Element Operator Network (eFEONet), specifically designed to address these challenges. eFEONet builds upon the FEONet framework (Lee et al., 2025), a highly data-efficient operator learning method that requires minimal training data or no dataset at all. Unlike traditional operator networks, eFEONet leverages the structure of finite element methods (FEMs), where the solution is expressed as a linear combination of nodal coefficients and basis functions. This design not only eliminates the need for large datasets but also ensures the exact satisfaction of boundary conditions. By incorporating insights from singular perturbation analysis in PDE theory, we design special basis functions within the finite element framework that capture the asymptotic behavior of solutions in boundary or interior layers (Gie et al., 2018). This approach enables accurate modeling of sharp transitions while

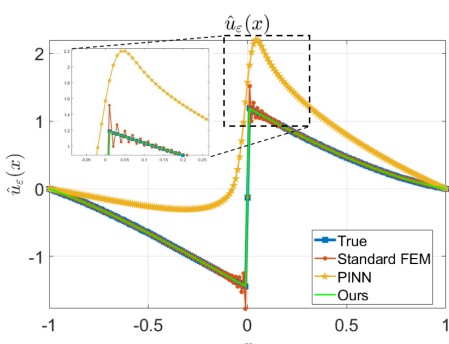

Figure 2: Comparison of the reference solution (True) and the predicted solutions using Standard FEM, PINN, and eFEONet (Ours) for the interior layer problem with $\varepsilon = 10^{-5}$.

maintaining computational efficiency. Recently, Component Fourier Neural Operator (ComFNO) (Li et al., 2024), a modified version of FNO, attempted to incorporate asymptotic behavior into its methodology. However, these approaches did not rigorously consider singular perturbation analysis, leading to suboptimal accuracy in capturing sharp layer structures. Moreover, as we can see in Section 4, despite utilizing 900 training samples, ComFNO exhibited an error that was two orders of magnitude higher than our eFEONet, which required no training data at all. We validate our approach through theoretical convergence analysis and empirical results on various singularly perturbed problems, including both boundary and interior layers (Chekroun et al., 2020; Gie et al., 2018). The results demonstrate that eFEONet achieves high accuracy and efficiency, even for problems with strong boundary or interior layer phenomena such as convection-dominated PDEs (Stynes, 2005)

The main contributions of the paper are summarized as follows:

- We propose eFEONet, which integrates singular perturbation analysis into the FEONet framework. This incorporation enables superior accuracy in solving singularly perturbed PDEs, effectively capturing sharp transitions in both boundary and interior layers.

- Singular perturbation problems typically require increasingly finer meshes as the parameter $\varepsilon > 0$ decreases, making dataset generation computationally very expensive. Our approach overcomes this limitation by being highly data-efficient, requiring minimal training data, or even operating without any training dataset.

- We demonstrate the effectiveness of eFEONet through comprehensive experiments on challenging convection-diffusion PDEs, including problems with boundary and interior layers in both 1D and 2D. The results show that eFEONet achieves error reductions of two orders of magnitude compared to existing approaches, even when no training data is used.

- The proposed eFEONet is supported by rigorous theoretical foundations through finite element approximation and asymptotic analysis. This robust framework enables a formal convergence analysis, ensuring both reliability and explainability.

## 2 RELATED WORKS

**Neural Operators.**  Operator learning trains models to approximate PDE solution operators using datasets of input-output pairs from numerical solvers (Bhatnagar et al., 2019; Guo et al., 2016; Khoo et al., 2017; Zhu & Zabaras, 2018), enabling efficient and real-time predictions for varying inputs (Li et al., 2020). Notable architectures include the Fourier Neural Operator (FNO) (Kovachki et al., 2021) and DeepONet (Lu et al., 2021a). Recent advances also explore message-passing frameworks to accommodate complex problem structures (Brandstetter et al., 2022; Lienen & Günnemann, 2022; Pfaff et al., 2021; Boussif et al., 2022). In addition, transformer-based architectures have been introduced (Cao, 2021; Wang et al., 2025; Hao et al., 2024), along with emerging foundation models tailored for PDEs(Herde et al., 2024; Ye et al., 2024). Despite these developments, operator learning still faces challenges in generalization, data efficiency, and resolving sharp solution features.

**Unsupervised Physics-based Operator Networks.**  Unsupervised physics-based operator networks incorporate governing equations directly into neural operator architectures, minimizing or completely removing the need for labeled training data. Variational frameworks such as FEONet (Lee et al., 2025) and SCLON (Choi et al., 2024) use PDE residuals in weak form to achieve accurate predictions without explicit simulation data. Similarly, physics-informed neural operator approaches like PINNs (Lu et al., 2021b; Han et al., 2018), PINO (Li et al., 2021b), and PIDeepONet (Wang et al., 2021) can also be formulated to rely entirely on PDE constraints and boundary conditions. Despite recent progress, accurately capturing multiscale phenomena and sharp gradients without labeled data remains challenging, highlighting the need for more robust unsupervised approaches.

**Neural networks for boundary layers**  Deep learning has emerged as a promising approach for solving singularly perturbed PDEs, with physics-informed methods also contributing to this effort (Arzani et al., 2023; Tawfiq & Al-Abrahemee, 2014). However, these approaches often lack scalability and remain effective only in limited scenarios. The study on stiff chemical kinetics (Goswami et al., 2024) utilizes deep neural operators specifically for reaction-diffusion stiffness, limiting its applicability compared to our method, which addresses a broader class of singularly perturbed PDEs, including boundary and interior layers, particularly in data-scarce scenarios. Recently, a homotopy-based approach to learn the singularly perturbed problems was proposed by CHEN et al. (2025) for specific PDE instances rather than operator learning approaches. ComFNO (Li et al., 2024) incorporates asymptotic expansions to better handle singular perturbations. Nonetheless, challenges persist, including the need for large training datasets, difficulty in accurately capturing sharp transitions, and a lack of rigorous theoretical foundations to ensure broader reliability.

## 3 ENRICHED FEONET

In this section, we shall describe our proposed method, eFEONet, designed for solving singularly perturbed parametric PDEs. We start by giving a brief overview of the FEMs, which form the core of our approach. Then, we will explain eFEONet, the main method we propose in this paper.

For the description, we will focus on the following PDE:

$$-\varepsilon \operatorname{div}\left(\boldsymbol{a}(\mathbf{x})\nabla u_{\varepsilon}\right) + \boldsymbol{b}(\mathbf{x}) \cdot \nabla u_{\varepsilon} = f \quad \text{in } D.$$
$$u_{\varepsilon} = 0 \quad \text{on } \partial D. \tag{1}$$

Here we assume that the singular perturbation parameter $\varepsilon > 0$ is very small so that the boundary layer phenomenon occurs. Furthermore, to highlight that the shape of a solution depends on $\varepsilon > 0$, we will denote the solution as $u_{\varepsilon}$.

As we will explain in more detail later, we propose an operator-learning approach for the singular perturbation problem that enables real-time solution predictions whenever the input data of the PDE varies. As a prototype model, we set the external force $f$ as an input of neural networks, and train the model so that the neural networks can learn the operator $\mathcal{G} : f \mapsto u_{\varepsilon}$. Note, however, that our method

can be easily extended to various forms of input functions, including boundary conditions, variable coefficients, or initial conditions (see, e.g., (Lee et al., 2025)).

## 3.1 Finite Element Method

The finite element method (FEM) is a technique for the numerical solution of PDEs and is based on the weak formulation of the PDE equation 1, which seeks to find a function $u_\varepsilon \in V$ satisfying

$$B[u_\varepsilon, v] := \varepsilon \int_D \boldsymbol{a}(\mathbf{x}) \nabla u_\varepsilon \cdot \nabla v \, \mathrm{d}\mathbf{x} + \int_D \boldsymbol{b}(\mathbf{x}) \cdot \nabla u_\varepsilon v \, \mathrm{d}\mathbf{x} = \int_D f v \, \mathrm{d}\mathbf{x} =: \ell(v) \quad \text{for all } v \in V, \quad (2)$$

where $V$ is typically an infinite-dimensional function space for the solution and test functions. The first step in finite element method (FEM) theory is to discretize the domain $D \subset \mathbb{R}^d$, known as a *triangulation*. For $d = 1$ and $D = [a, b]$, this involves points $a = x_0 < x_1 < \cdots < x_K = b$, with each interval $[x_{i-1}, x_i]$ forming a 1-simplex. For $d = 2$, the triangulation consists of closed triangles $T_i$ (2-simplexes), $i = 1, \ldots, K$, whose interiors are disjoint. If $i \neq j$ and $T_i \cap T_j \neq \emptyset$, then the intersection is either a shared vertex or edge. For $d \geq 3$, elements are $d$-simplexes. Let $h_T$ denote the longest edge of a triangle $T$, and define the global mesh size as $h = \max_T h_T$. Let $S_h$ be the space of continuous functions $v_h$ on $D$ such that the restriction of $v_h$ to each element is a polynomial. The finite-dimensional ansatz space is then defined as $V_h = S_h \cap V$. Let $\{\mathbf{x}_i\}$ denote the triangulation vertices, and $\{\phi_j\}$ the *nodal basis* for $V_h$, where $\phi_j(\mathbf{x}_i) = \delta_{ij}$. Using piecewise linear basis functions defines the P1-element method; using piecewise quadratic polynomials gives the P2-element method. The dimension of $V_h$ depends on the triangulation and hence on the mesh parameter $h$.

The FEM aims to approximate the infinite-dimensional space $V$ by a finite-dimensional subspace $V_h$ defined by $V_h = \mathrm{span}\{\phi_1, \phi_2, \cdots, \phi_{N(h)}\}$. This makes the problem numerically solvable. Motivated from equation 2, we seek to compute the approximate solution $u_{\varepsilon,h} \in V_h$ using the so-called *Galerkin approximation*, which is given by the equation $B[u_{\varepsilon,h}, v_h] = \ell(v_h)$ for all $v_h \in V_h$. Writing the finite element solution as $u_{\varepsilon,h}(\mathbf{x}) = \sum_{k=1}^{N(h)} \alpha_k \phi_k(\mathbf{x})$ with $\alpha_i \in \mathbb{R}$, the Galerkin approximation transforms into the linear algebraic system $A\alpha = F$ with $A_{ik} := B[\phi_k, \phi_i]$ and $F_i := \ell(\phi_i)$. Here, $A$ is invertible, assuming the underlying PDE has an appropriate structure. The coefficients $\{\alpha_k\}_{k=1}^{N(h)}$ is then determined, thus yielding the approximate solution $u_{\varepsilon,h}$.

## 3.2 Enriched FEONet with a Corrector Basis

Now we are ready to introduce our main method, the enriched FEONet (eFEONet). One key novelty of the eFEONet is to utilize extra basis functions derived from theoretical arguments (see, e.g., Appendix B). For a clear illustration of the proposed method, we shall explain it through a simple example of the following form:

$$\begin{aligned} -\varepsilon u_\varepsilon'' - u_\varepsilon' &= f(x), \quad x \in (-1, 1), \\ u_\varepsilon(-1) &= u_\varepsilon(1) = 0, \end{aligned} \quad (3)$$

where $0 < \varepsilon \ll 1$. As we can see from Figure 2, when $\varepsilon > 0$ is small, it is difficult to expect the classical FEM or the original FEONet to achieve good performance due to the sharp transitions near the boundary. To accurately capture the boundary layer, we incorporate an additional basis function, commonly referred to as the *corrector function* in mathematical analysis, for example in this case, defined as: $\phi_{\mathrm{cor}}(x) := e^{-(1+x)/\varepsilon} - (1 - (1 - e^{-2/\varepsilon})(x + 1)/2)$. Such a basis function reflects the boundary layer properties of the given equation and is derived from theoretical arguments. The derivation of various corrector basis functions will be addressed in Appendix B. The corrector basis is added to the standard nodal basis functions of FEM to construct an enriched Galerkin space. In other words, for enriched FEONet for the singularly-perturbed problems, we now replace the original ansatz space $V_h$ by the enriched Galerkin space $\overline{V}_h = \{\phi_{\mathrm{cor}}, \phi_1, \phi_2, \cdots, \phi_{N(h)}\}$, where the corrector basis $\phi_{\mathrm{cor}}$ has been added to $V_h$. It is noteworthy that no significant additional computational cost occurs, as the enriched basis is only restricted to boundary elements. In general, neural networks assume a smooth prior, which makes them less effective in handling boundary layers. This can lead to unstable training due to the direct calculation of the PDE residual. In contrast, the eFEONet leverages theory-guided basis functions, allowing its predicted solution to precisely capture the sharp transitions near the boundary. Encapsulating the above discussion, the enriched FEM for the boundary layer problem can be written as follows: we seek $u_{\varepsilon,h}^{\mathrm{en}} \in \overline{V}_h$ satisfying

$$B[u_{\varepsilon,h}^{\mathrm{en}}, v_h] = \ell(v_h) \quad \text{for all } v_h \in \overline{V}_h. \quad (4)$$

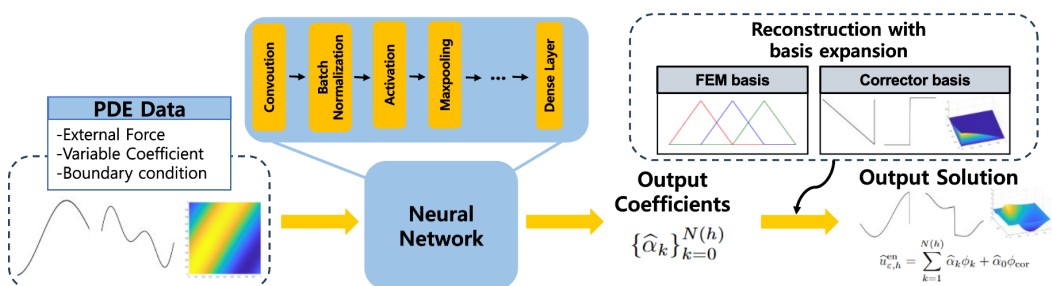

Figure 3: Schematic illustration of eFEONet.

In our eFEONet approach, the input to the neural network consists of data related to the given PDE problems, which is parameterized by $\boldsymbol{\omega} \in \Omega$, while the output consists of the coefficients of a basis expansion. To be more specific, we incorporate this into a deep learning framework to construct the eFEONet, whose solution prediction is written as

$$\widehat{u}_{\varepsilon,h}^{\mathrm{en}}(\mathbf{x}; \boldsymbol{\omega}) = \sum_{k=1}^{N(h)} \widehat{\alpha}_k(\boldsymbol{\omega}) \phi_k(\mathbf{x}) + \widehat{\alpha}_0(\boldsymbol{\omega}) \phi_{\mathrm{cor}}, \tag{5}$$

where the dimension of the output of the neural network has increased by one to handle the added corrector basis. By writing $\phi_0 := \phi_{\mathrm{cor}}$, the loss function for the eFEONet is defined as

$$\mathcal{L}^M(\widehat{u}_{\varepsilon,h}^{\mathrm{en}}) = \frac{1}{M} \sum_{m=1}^{M} \sum_{i=0}^{N(h)} |B[\widehat{u}_{\varepsilon,h}^{\mathrm{en}}(\mathbf{x}; \boldsymbol{\omega}_m), \phi_i(\mathbf{x})] - \ell(\phi_i(\mathbf{x}); \boldsymbol{\omega}_m)|^2, \tag{6}$$

for randomly drawn parameters $\boldsymbol{\omega}_1, \cdots, \boldsymbol{\omega}_M \in \Omega$. A schematic diagram of the eFEONet algorithm is shown in Figure 3.

*Remark* 3.1. Our framework employs corrector functions tailored to specific problem classes, yet they are not confined to individual instances. For families of PDEs with analogous singular behavior, the same correctors can often be applied effectively. In convection–diffusion equations, for example, the boundary layer typically has a thickness proportional to $\varepsilon$ with an exponential profile, a structure preserved even with additional reaction terms.

*Remark* 3.2. Some preliminary results show that one could attempt to learn the corrector bases using data (see, e.g., Appendix E.5). In contrast, our approach constructs them via classical numerical analysis, which not only requires no data but also achieves substantially better performance. This integration of analytical methods into an operator learning framework constitutes the main novelty of our work, highlighting how analytic knowledge can maximize the efficiency of operator learning.

### 3.3 Convergence of Enriched FEONet

In this section, we investigate the convergence result for eFEONet, providing a theoretical foundation for the proposed approach. Let us denote the solution of equation 3 by $u_\varepsilon$ corresponding to a given parameter $0 < \varepsilon \ll 1$. Since our method is built upon the enriched FEM, the enriched finite element approximation $u_{\varepsilon,h}^{\mathrm{en}}$ in equation 4 serves as an intermediate step between the exact solution $u_\varepsilon$ and the approximate solution $\widehat{u}_{\varepsilon,h}^{\mathrm{en}}$ obtained from eFEONet equation 5. Therefore, for the purpose of error analysis, the error $u_\varepsilon - \widehat{u}_{\varepsilon,h}^{\mathrm{en}}$ is decomposed into two components, specifically $u_\varepsilon - \widehat{u}_{\varepsilon,h}^{\mathrm{en}} = (u_\varepsilon - u_{\varepsilon,h}^{\mathrm{en}}) + (u_{\varepsilon,h}^{\mathrm{en}} - \widehat{u}_{\varepsilon,h}^{\mathrm{en}}) =: \text{(I)} + \text{(II)}$. The error analysis for the first term (I) is well investigated in the previous literature on singular perturbation analysis. For example, in (Cheng & Temam, 2002), the following error estimate was derived for the enriched FEM equation 4:

$$\|u_\varepsilon - u_{\varepsilon,h}^{\mathrm{en}}\|_{H^1} \le C \left( h + \frac{h^2}{\varepsilon} \right), \tag{7}$$

where $C > 0$ is a constant independent of $h$ and $\varepsilon$. This result is especially highlighted as it provides a satisfactory convergence result even in the under-resolved case for $h > \varepsilon$. More general results can be found in various papers, e.g., from (Jung, 2005; Gie et al., 2018). The second error (II) represents

Table 1: Mean relative $L^2$ test errors ($\times 10^{-3}$) for FNO, ComFNO, DeepONet, PINN, and eFEONet by varying the number of training input-output data pairs. Here, we set $\varepsilon = 10^{-3}$ for all experiments.

| Model | Exp1. ODE w/ boundary layer | | | | Exp2. ODE w/ interior layer | | | | Exp3. PDE on square | | | |
|---|---|---|---|---|---|---|---|---|---|---|---|---|
| | # of training data | | | | # of training data | | | | # of training data | | | |
| | 900 | 90 | 9 | None | 900 | 90 | 9 | None | 900 | 90 | 9 | None |
| FNO | 36.0 | 68.3 | 382 | - | 84.2 | 153 | 961 | - | 10.3 | 1e+03 | 1e+05 | - |
| ComFNO | 3.88 | 51.1 | 347 | - | 8.21 | 126 | 876 | - | 15.1 | 1320 | 1e+05 | - |
| DeepONet | 23.9 | 101 | 286 | - | 7.40 | 6.80 | 240 | - | 2300 | 1780 | 1590 | - |
| PINN | - | - | - | 970 | - | - | - | 800 | - | - | - | 1174 |
| eFEONet (Ours) | **0.01** | **0.07** | **0.03** | **0.06** | **1.79** | **1.99** | **4.23** | **3.17** | **2.26** | **1.83** | **5.38** | **8.53** |

a critical aspect that requires novel theoretical analysis. The parameters of $\widehat{u}_{\varepsilon,h}^{\mathrm{en}} =: \widehat{u}_{\varepsilon,h,n,M}^{\mathrm{en}}$ relevant to convergence include the neural network architecture and the number of sampling points $\boldsymbol{\omega} \in \Omega$, denoted by $n \in \mathbb{N}$ and $M \in \mathbb{N}$, respectively. A larger $n \in \mathbb{N}$ indicates a greater approximation capacity of the neural networks. We will establish the following theorem addressing the second error (II), ensuring the reliability of our method and providing the theoretical underpinning of the proposed numerical scheme. The detailed mathematical formulation and proof are presented in Theorem C.7.

**Theorem 3.3.** *Let $u_{\varepsilon,h}^{\mathrm{en}}$ be the enriched finite element approximation equation 4 of the true solution $u_\varepsilon$ and $\widehat{u}_{\varepsilon,h,n,M}^{\mathrm{en}}$ be the approximate solution computed by the eFEONet. Then there holds*

$$\mathbb{E}\left[\|u_{\varepsilon,h} - \widehat{u}_{\varepsilon,h,n,M}^{\mathrm{en}}\|_{L^2(D)}^2\right] \to 0 \quad \text{as } n, M \to \infty, \tag{8}$$

*where the expectation is taken over random samples $\boldsymbol{\omega} \in \Omega$.*

*Remark* 3.4. The main difference in our convergence analysis from the original FEONet lies in the singular perturbation analysis, which governs equation 7, while the approximation and generalization errors retain a similar structure. Unlike FEONet, however, the associated constants in our setting depend implicitly on the perturbation parameter $\varepsilon$. Making this dependence explicit and establishing a unified error estimate in terms of the mesh size $h$, perturbation parameter $\varepsilon$, approximation parameter $n$, and number of samples $M$ remains an interesting research direction to be addressed in future work.

## 4 EXPERIMENTS

In this section, we evaluate the performance of eFEONet on four distinct types of singularly perturbed differential equations, including both ordinary and partial differential equations. For ordinary differential equations (ODEs), we examine scenarios with and without turning points, highlighting eFEONet's adaptability to varying problems. For PDEs, we test the eFEONet over domains with square and circle geometries to assess its robustness across different spatial configurations. Furthermore, we conduct a comparison of the experimental results with those obtained using FNO (Kovachki et al., 2021) and ComFNO (Li et al., 2024), a neural operator model specifically designed to address the challenges of singularly perturbed differential equations.

The high-precision numerical solutions are denoted as $u_\varepsilon$, while the predictions are represented as $\widehat{u}_\varepsilon$. The training dataset consists of 900 load vectors generated from independently sampled functions $f$, with inputs discretized at a resolution of 201 for both 1D and 2D cases (see Appendix D.1). High-precision numerical solutions on the Shishkin mesh (see, e.g., (Li et al., 2024)) are used to compute the corresponding outputs $u_\varepsilon$, which serve as the ground truth during training. Additionally, for all ODE experiments, the input-output resolution is set to 201, ensuring consistency across the comparative evaluations of FNO, ComFNO, and our method. In 2D PDE experiments, the resolution is fixed at 51 for $\varepsilon = 10^{-3}$ and for $\varepsilon = 10^{-4}$ in the rectangular domain. For the circle domain, an input-output resolution of 960 is used for the irregular geometry.

### 4.1 ORDINARY DIFFERENTIAL EQUATIONS WITH BOUNDARY LAYER

We begin with the following problem:

$$-\varepsilon u_\varepsilon'' + (x+1)u_\varepsilon' = f(x), x \in (0,1),$$
$$u_\varepsilon(0) = u_\varepsilon(1) = 0. \tag{9}$$

Table 2: Mean relative $L^2$ test errors($\times \mathbf{10^{-3}}$) for FNO, ComFNO, and eFEONet across different values of $\varepsilon$ for ODEs with boundary layers. FNO and ComFNO are trained with 900 samples, whereas eFEONet uses no pre-computed training data.

| Model | Varying $\varepsilon$ | | | |
|---|---|---|---|---|
| | $\varepsilon = 10^{-3}$ | $\varepsilon = 10^{-4}$ | $\varepsilon = 10^{-5}$ | $\varepsilon = 10^{-6}$ |
| FNO (w/ 900 train data) | 36 | 36.8 | 36.9 | 36.9 |
| ComFNO (w/ 900 train data) | 3.88 | 5.7 | 7.60 | 5.66 |
| Standard FEM | 98.1 | 382 | 3.04e+03 | 6.80e+03 |
| Ours (eFEONet) (w/o train data) | **0.07** | **0.03** | **0.07** | **0.03** |

As shown in Figure 4, the solution exhibits an exponential boundary layer near $x = 1$, making it an excellent test case for evaluating the ability of eFEONet to capture sharp boundary layers effectively. To address this challenge for equation 9, eFEONet utilizes the corrector $\phi_0(x) = \exp(-2(1-x)/\varepsilon)$ to capture the boundary layer more effectively (see Appendix D for further details).

As shown in the second column of Table 1, when sufficient training data is available, both FNO and ComFNO achieved reasonable accuracy, but our eFEONet outperforms them. Moreover, as the amount of training data decreases, the error for ComFNO increases significantly, whereas eFEONet maintains higher accuracy even with limited data. Table 2 presents the relative $L^2$ test errors for FNO, ComFNO, and eFEONet across different values of $\varepsilon$. The results demonstrate that eFEONet consistently outperforms the benchmark models, achieving significantly lower errors even without using any training

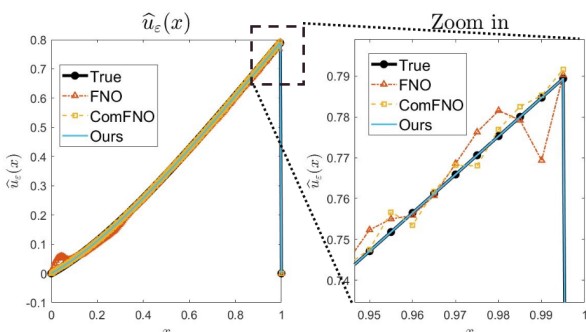

Figure 4: Comparison of predicted solutions $\widehat{u}_\varepsilon$ using FNO, ComFNO, and eFEONet with $\varepsilon = 10^{-4}$ for the boundary layer problem. The external force input function is given by $f(x) = 1.81 \sin(1.68x) + 0.09 \cos(-1.78x)$.

data. Note that the error trends for FNO and ComFNO remain relatively stable across different $\varepsilon$ values, but eFEONet maintains even higher accuracy across all tested cases, demonstrating its effectiveness in capturing boundary layer phenomena without requiring extensive training datasets.

Figure 4 further compares the predicted solution $\widehat{u}_\varepsilon$ for one of the test samples using FNO, ComFNO, and eFEONet with $\varepsilon = 10^{-4}$. FNO shows substantial errors, particularly near the boundary layer, while ComFNO achieves relatively better accuracy but struggles to fully resolve the sharp transitions. In contrast, eFEONet, leveraging the corrector function as an additional basis function, achieves the highest accuracy, effectively capturing the boundary layer with minimal error.

### 4.2 Ordinary Differential Equations with Interior Layer

We consider the following ordinary differential equation with a turning point at $x = 0$:

$$-\varepsilon u''_\varepsilon - xu'_\varepsilon = f(x), \quad x \in (-1, 1),$$
$$u_\varepsilon(-1) = u_\varepsilon(1) = 0,$$

(10)

with the corrector function $\phi_0(x) = \text{erf}(\sqrt{1/(2\varepsilon)}x)$. As shown in the third column of Table 1, eFEONet achieves better accuracy than both FNO and ComFNO, with a larger performance gap emerging as the number of training samples decreases. This highlights the robustness of eFEONet in data-scarce scenarios. Table 3 shows the relative $L^2$ test errors for FNO, ComFNO, and eFEONet across different values of $\varepsilon$ for ODEs with interior layers. The results demonstrate that eFEONet consistently achieves superior accuracy compared to FNO and ComFNO, even in the absence of training data. Notably, as $\varepsilon$ decreases, the performance gap between eFEONet and the benchmark models significantly widens, indicating eFEONet's ability to accurately capture sharp interior layers.

Figure 5 compares the predicted solutions $\widehat{u}_\varepsilon$ for two test samples using FNO, ComFNO, and eFEONet with $\varepsilon = 10^{-8}$. Notably, eFEONet demonstrates superior accuracy, particularly around the

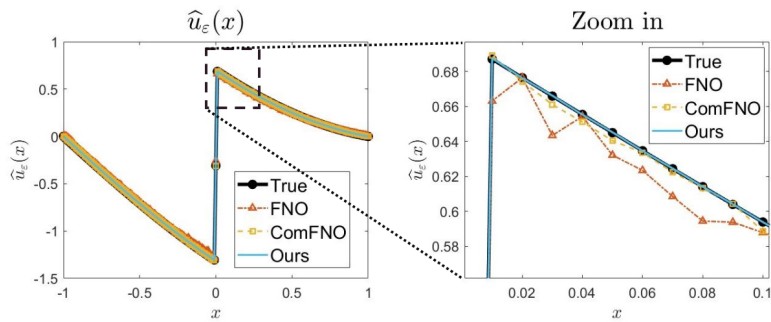

Figure 5: Comparison of predicted solutions $\widehat{u}_\varepsilon$ using FNO, ComFNO, and eFEONet with $\varepsilon = 10^{-8}$. The external forcing input is given by $f(x) = x(-0.58\sin(0.44x) + 1.61\cos(1.05x))$.

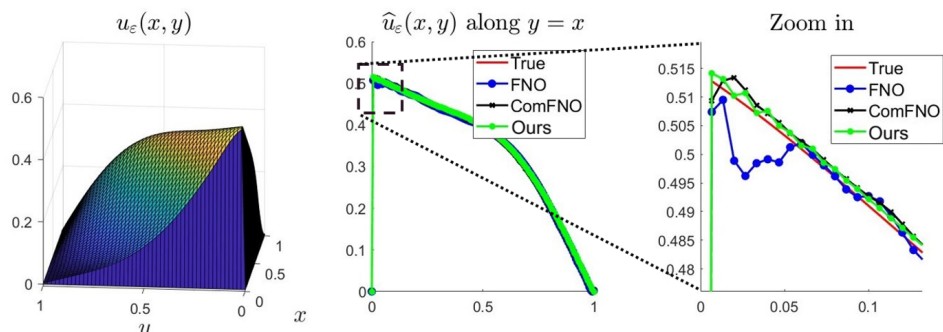

Figure 6: Comparison of the reference solution $u_\varepsilon(x, y)$ (left) and the predicted solutions $\widehat{u}_\varepsilon$ along the diagonal $y = x$ for ComFNO and eFEONet (middle and right) with $\varepsilon = 10^{-4}$. The results highlight the superior accuracy of eFEONet in capturing sharp boundary layers along $x = 0$, whereas ComFNO exhibits noticeable errors near the boundary regions.

singular region near the turning point at $x = 0$. This result underscores the capability of eFEONet to effectively handle the challenges posed by singularities and turning points in differential equations, delivering reliable predictions even in complex scenarios.

### 4.3 PARTIAL DIFFERENTIAL EQUATIONS ON SQUARE

For a boundary-value problem of an elliptic PDE in the spatial domain $D = [0, 1]^2$, we consider

$$-\varepsilon\Delta u_\varepsilon - (1, 1) \cdot \nabla u_\varepsilon = f(x, y) \text{ in } D,$$
$$u_\varepsilon(x, y) = 0 \text{ on } \partial D, \quad (11)$$

where the solution exhibits a boundary layer along the edge at $x = 0$ and $y = 0$, as illustrated in Figure 14.

Table 3: Mean relative $L^2$ test errors($\times \mathbf{10^{-3}}$) for FNO, ComFNO, and eFEONet across different values of $\varepsilon$ for ODEs with interior layers. The results highlight the performance of each model when trained with 900 data samples (FNO, ComFNO) and without training data (eFEONet).

| Model | Varying $\varepsilon$ | | | |
|---|---|---|---|---|
| | $\varepsilon = 10^{-3}$ | $\varepsilon = 10^{-4}$ | $\varepsilon = 10^{-5}$ | $\varepsilon = 10^{-6}$ |
| FNOw/ 900 train data | 84.2 | 86.9 | 81.5 | 86.9 |
| ComFNOw/ 900 train data | 8.21 | 8.97 | 19.6 | 15.5 |
| Standard FEMw/o train data | **0.74** | **4.3** | 22.2 | 75.4 |
| Ours(eFEONet)w/o train data | 3.17 | 5.21 | **0.66** | **0.19** |

As shown in the fourth column of Table 1, the accuracy gap between eFEONet and benchmark models becomes even more pronounced for this problem. This highlights the capability of eFEONet to effectively resolve boundary layers in complex spatial domains. Furthermore, as seen in Figure 6, ComFNO shows large errors, whereas eFEONet achieves consistently low errors across the entire domain, demonstrating its robustness and superior accuracy in handling such challenging scenarios.

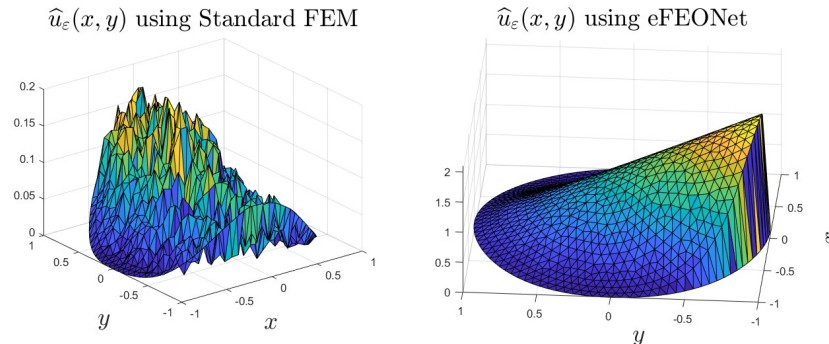

Figure 7: Numerical results comparing standard FEM (left) and eFEONet with a corrector function (right) for $\varepsilon = 10^{-3}$. The standard FEM solution exhibits significant numerical oscillations, failing to capture the sharp transition accurately, whereas eFEONet effectively approximates the true solution.

### 4.4 PARTIAL DIFFERENTIAL EQUATIONS ON CIRCLE

We consider a singularly perturbed differential equation in a circular domain $D$, which is a circle centered at $(0, 0)$ with a radius of 1, given by

$$
\begin{aligned}
-\varepsilon \Delta u_\varepsilon - (u_\varepsilon)_y &= f(x, y) && \text{in } D, \\
u_\varepsilon(x, y) &= 0 && \text{on } \partial D.
\end{aligned}
\tag{12}
$$

A boundary layer forms only along the outflow boundary, which corresponds to the lower semicircle. Moreover, the boundary layer thickness is non-uniform, becoming thicker near $(\pm 1, 0)$. These properties make the problem analytically challenging. Hence, conventional numerical methods struggle to handle this problem, necessitating a scheme based on singular perturbation analysis to achieve accurate solutions (Gie et al., 2018; Hong et al., 2014). The corrector function is given by

$$
\theta^0(\eta, \xi) = -u^0(\cos \eta, \sin \eta) \exp \left( \frac{\sin \eta}{\varepsilon} \xi \right) \chi_{[\pi, 2\pi]}(\eta),
$$

where $\eta$ represents the tangential direction, $\xi$ denotes the normal direction in a boundary-fitted coordinate system and $\chi$ is the characteristic function. From the form of this corrector, it is evident that the boundary layer thickness varies with $\eta$ and exhibits degeneracy at $(\pm 1, 0)$. Specifically, the exponential term shows that the thickness is proportional to $\sin \eta$, meaning that it becomes significantly larger near $\eta = \pm \pi/2$ (corresponding to $(\pm 1, 0)$). This non-uniform behavior complicates the analysis and requires a careful treatment of the singular perturbation structure.

As shown in Figure 16 and Figure 17, the sharp transition occurs near $(0, -1)$. Problems with non-square geometries pose a significant challenge for existing neural operator models in learning the solution operator. However, eFEONet demonstrates robust performance, even in such complex geometry settings, achieving high accuracy even with limited training data as shown in Figure 7.

## 5 CONCLUSION AND LIMITATIONS

In this paper, we introduced eFEONet, designed for singularly perturbed differential equations. By integrating boundary layer theory into the finite element framework, eFEONet captures sharp transitions using theory-guided basis functions, eliminating the need for extensive training datasets. Experimental results demonstrate the robustness of eFEONet across various PDEs with boundary and interior layers in different geometries. Compared to FNO and ComFNO, eFEONet consistently achieves superior accuracy, particularly in data-scarce scenarios. Additionally, our method is supported by convergence analysis, validating its reliability. Despite its strong performance, certain limitations remain. First, the choice of parameters, such as the number of basis functions and network hyperparameters, significantly affects the learning dynamics and overall performance of eFEONet. A systematic analysis of these parameters is still an open research question. Second, while our study presents a unique method for solving singularly perturbed problems with boundary and interior layers using minimal or even no training data, future research should extend eFEONet to handle more challenging problems, such as corner singularities and other intricate geometrical effects.

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

## A    USE OF LARGE LANGUAGE MODELS (LLMs)

We acknowledge the use of a large language model (OpenAI ChatGPT) as a general-purpose assist tool during the preparation of this work. The LLM was used to improve clarity, grammar, and style in the abstract, introduction, and related work sections. The authors take full responsibility for all scientific content presented in this paper.

## B    DERIVATION OF CORRECTOR BASIS FUNCTIONS

We have focused on the numerical treatment of the following singularly perturbed convection-dominated problem

$$-\varepsilon \Delta u_\varepsilon - \boldsymbol{b} \cdot \nabla u_\varepsilon + c u_\varepsilon = f \quad \text{in } D,$$
$$u_\varepsilon = 0 \quad \text{on } \partial D,$$

where $0 < \varepsilon \ll 1$, and $\boldsymbol{b} = \boldsymbol{b}(\mathbf{x})$, $c = c(\mathbf{x})$ and $f = f(\mathbf{x})$ are given smooth functions defined over the domain $D$. This formulation represents a general convection-diffusion-reaction equation with singular perturbation. For this problem, we considered both 1D and 2D settings, addressing critical challenges such as boundary layers and interior layers that arise due to the small parameter $\varepsilon > 0$. From this point onward, our analysis follows the singular perturbation analysis stated in (Gie et al., 2018). The theoretical foundations and techniques presented here are based on this approach, providing a rigorous framework for handling boundary and interior layers in singularly perturbed problems. For further details on related studies and extensions, we refer the reader to (Gie et al., 2018).

**(Boundary layer case)** While our ultimate goal is to solve the above problem in 2D, we first simplify the analysis and explanation by considering a one-dimensional paradigm problem. The 1D problem is defined as

$$-\varepsilon u''_\varepsilon - u'_\varepsilon = f \quad \text{in } (0,1),$$
$$u_\varepsilon(0) = u_\varepsilon(1) = 0.$$

This 1D model provides a clear framework for understanding boundary layer phenomena and allows us to systematically develop the necessary mathematical and computational tools before extending the approach to higher dimensions. The corresponding limit problem is obtained by formally setting $\varepsilon = 0$:

$$-u'_0 = f \quad \text{in } (0,1),$$
$$u_0(1) = 0.$$

Treating this as a transport equation, we supplement the limit problem with the inflow boundary condition at $x = 1$, namely

$$u_0(1) = 0.$$

Solving this equation with the given condition yields

$$u_0 = -\int_x^1 f(s)\, \mathrm{d}s.$$

At this stage, the choice of the inflow boundary condition $u_0(1) = 0$ is an assumption motivated by the structure of the transport equation. To address the boundary layer near $x = 0$, we introduce a stretched variable $\bar{x} = x/\varepsilon^\alpha$, with $\alpha > 0$. Substituting $\bar{x}$ into the original problem with $f = 0$, we derive

$$-\varepsilon^{1-2\alpha} \frac{\mathrm{d}^2 u_\varepsilon}{\mathrm{d}\bar{x}^2} - \varepsilon^{-\alpha} \frac{\mathrm{d}u_\varepsilon}{\mathrm{d}\bar{x}} = 0.$$

Here, $f$ is omitted because it is accounted for in the inviscid equation $-u'_0 = f$. To define a corrector from this equation, we observe that the corrector must balance the difference between $u_\varepsilon$ and $u_0$ at $x = 0$ and decay rapidly as $x$ moves away from 0. By setting $1 - 2\alpha = -\alpha$, we find $\alpha = 1$, resulting in the following boundary layer equation

$$-\frac{\mathrm{d}^2 \bar{\theta}_\varepsilon}{\mathrm{d}\bar{x}^2} - \frac{\mathrm{d}\bar{\theta}_\varepsilon}{\mathrm{d}\bar{x}} = 0.$$

The boundary conditions for this equation are

$$\bar{\theta}_\varepsilon(0) = -u_0(0), \quad \bar{\theta}_\varepsilon \to 0 \quad \text{as } \bar{x} \to \infty.$$

The explicit solution for $\bar{\theta}_\varepsilon$, the approximate corrector, is given as

$$\bar{\theta}_\varepsilon = -u_0(0)e^{-\bar{x}} = -u_0(0)e^{-x/\varepsilon}.$$

As discussed earlier, we want to add this boundary layer function into our finite element ansatz space. However, note that this boundary layer function does not satisfy the appropriate boundary conditions. This is easily handled by introducing the boundary layer basis function of the form

$$\phi_0(x) = e^{-x/\varepsilon} + (1 - e^{-1/\varepsilon})x + 1.$$

**(Interior layer case):** For convection–diffusion equations with an interior layer, we consider the problem

$$-\varepsilon\, u_\varepsilon'' - b(x)u_\varepsilon' = f \quad \text{in } (-1, 1),$$
$$u_\varepsilon(-1) = u_\varepsilon(1) = 0,$$

where $b(x)$ satisfies $b < 0$ for $x < 0$, $b(0) = 0$, $b > 0$ for $x > 0$, and $b'(x) > 0$. The turning point at $x = 0$ introduces an interior layer due to the change in sign of $b(x)$, where characteristics collide. For the formal limit problem, setting $\varepsilon = 0$ leads to:

$$-b(x)u_0' = f,$$

but this may not be well-defined at $x = 0$ since $b(0) = 0$. Therefore, we split the solution into left and right parts, $u_0^l$ and $u_0^r$, corresponding to $x < 0$ and $x > 0$, respectively

$$-b(x)(u_0^l)' = f \quad \text{for } x < 0 \quad \text{and} \quad -b(x)(u_0^r)' = f \quad \text{for } x > 0.$$

The inflow boundary conditions are then supplemented as

$$u_0^l(-1) = 0, \quad u_0^r(1) = 0.$$

The discrepancy at $x = 0$ between $u_0^l$ and $u_0^r$ produces an interior layer. If $f(0) = 0$, the correctors introduced below can effectively capture the sharpness of this layer. However, if $f(0) \neq 0$, the limit problem

$$-b(x)u_0' = f$$

has an inconsistency at $x = 0$ because $b(0) = 0$. This implies that $u_0'$ diverges near $x = 0$, and the interior layer cannot be fully captured by standard corrector functions. To address this issue, the data may need to be adjusted to ensure compatibility, as described in related perturbation analyses. To analyze the interior layer, we introduce the stretched variable $\bar{x} = x/\sqrt{\varepsilon}$ and approximate $b(x)$ as $b(x) = b'(0)x + \frac{1}{2}b''(\xi)x^2 \approx b'(0)\sqrt{\varepsilon}\bar{x}$. Substituting these into the original equation with $f = 0$, we obtain the leading-order differential equation

$$-\frac{\mathrm{d}^2\theta}{\mathrm{d}\bar{x}^2} - b'(0)\bar{x}\frac{\mathrm{d}\theta}{\mathrm{d}\bar{x}} = 0,$$

subject to the boundary conditions

$$\theta \to \text{constant as } \bar{x} \to \pm\infty.$$

The solution of this equation, written explicitly, is

$$\theta = \frac{2}{\sqrt{\pi}} \int_0^{\bar{x}\sqrt{b'(0)/2}} e^{-\tau^2}\, \mathrm{d}\tau = \text{erf}\left(\bar{x}\sqrt{\frac{b'(0)}{2}}\right) = \text{erf}\left(x\sqrt{\frac{b'(0)}{2\varepsilon}}\right),$$

where erf denotes an error function. This serves as a corrector for the interior layer.

**(2D circular domain case):** We now investigate singularly perturbed differential equations of the form

$$-\varepsilon\Delta u_\varepsilon - (u_\varepsilon)_y = f(x, y) \quad \text{in } D,$$
$$u_\varepsilon = 0 \quad \text{on } \partial D,$$

where $0 < \varepsilon \ll 1$, $D$ represents the unit disk centered at $(0,0)$, and $f(x,y)$ is a smooth function defined over $D$. To analyze the asymptotic behavior as $\varepsilon \to 0$, we consider the corresponding limit problem, obtained by formally setting $\varepsilon = 0$:

$$-u_y^0 = f(x,y) \quad \text{in } D,$$
$$u^0 = 0 \quad \text{on } \Gamma_u,$$

where $\Gamma_u$ is the upper semicircle, defined as $\Gamma_u = \{(x,y) \mid x^2 + y^2 = 1, y > 0\}$. While the equation itself is straightforward, the justification of the boundary condition on $\Gamma_u$ requires careful analysis. The explicit solution for $u^0(x,y)$ can be derived as

$$u^0(x,y) = \int_y^{C_u(x)} f(x,s)\,\mathrm{d}s, \quad (x,y) \in D,$$

where $C_u(x) = \sqrt{1 - x^2}$. Notably, if $f(x,y)$ does not vanish at the characteristic points $(\pm 1, 0)$, the solution becomes incompatible, leading to singularities in the derivatives of $u^0$. To ensure well-posedness, we impose compatibility conditions of the form

$$\frac{\partial^{p_1 + p_2} f}{\partial x^{p_1} \partial y^{p_2}} = 0 \quad \text{at} \quad (\pm 1, 0),$$

for non-negative integers $p_1, p_2$ with $0 \le 2p_1 + p_2 \le 2$. This ensures smoothness at the characteristic points and prevents singularities from arising in the derivatives of $u^0$.

To analyze the singularly perturbed problem, we introduce a boundary-fitted coordinate system defined as

$$x = (1 - \xi) \cos \eta, \quad y = (1 - \xi) \sin \eta,$$

where $\xi = 1 - r$ represents the normal distance to the boundary, and $\eta$ is the polar angle measured from the $x$-axis. Using this transformation, we redefine the computational domain in terms of $(\eta, \xi)$ coordinates

$$D^* = \{(\eta, \xi) \in (0, 2\pi) \times (0,1)\}, \quad D_{\frac{1}{2}} = \{(\eta, \xi) \in D^* : \xi \le \frac{1}{2}\}.$$

Applying this change of variables, the partial derivatives transform as follows

$$\frac{\partial}{\partial x} = -\cos\eta \frac{\partial}{\partial \xi} - \frac{\sin\eta}{1 - \xi}\frac{\partial}{\partial \eta}, \quad \frac{\partial}{\partial y} = -\sin\eta\frac{\partial}{\partial \xi} + \frac{\cos\eta}{1 - \xi}\frac{\partial}{\partial \eta}.$$

Rewriting the differential operator in terms of $(\eta, \xi)$, we obtain

$$-\varepsilon\Delta u_\varepsilon - (u_\varepsilon)_y, = -\frac{\varepsilon}{(1-\xi)^2}\frac{\partial^2 u_\varepsilon}{\partial \eta^2} + \frac{\varepsilon}{1-\xi}\frac{\partial u_\varepsilon}{\partial \xi} - \varepsilon\frac{\partial^2 u_\varepsilon}{\partial \xi^2} + \sin\eta\frac{\partial u_\varepsilon}{\partial \xi} - \frac{\cos\eta}{1-\xi}\frac{\partial u_\varepsilon}{\partial \eta}.$$

To systematically analyze the singular perturbation, we seek an asymptotic expansion of $u_\varepsilon$ in the form

$$u_\varepsilon \sim \sum_{j=0}^\infty \left( \varepsilon^j u^j + \varepsilon^j \theta^j \right),$$

where $u^j$ corresponds to the outer expansion (valid away from the boundary layers), and $\theta^j$ represents the inner expansion (boundary layer correction). First, considering only the outer expansion, we assume

$$u_\varepsilon \sim \sum_{j=0}^\infty \varepsilon^j u^j.$$

Substituting this expansion into the governing equation at each order of $\varepsilon$, we obtain the leading-order equation for all $j \ge 0$:

$$-u^j = \Delta u^{j-1} \quad \text{in } D,$$
$$u^j = 0 \quad \text{on } \Gamma_u.$$

Here, we set $\Delta u^{-1} = f(x,y)$ for simplicity. The justification of the boundary condition is nontrivial and follows from convergence theorems. To analyze the boundary layer, we introduce a stretched variable

$$\bar{\xi} = \xi^\alpha \bar{\xi},$$

where the appropriate choice of $\alpha$ will be determined by asymptotic analysis. Setting $f = 0$ in the original equation and substituting the new variables, we obtain the transformed equation

$$-\frac{\varepsilon}{(1 - \varepsilon^\alpha \bar{\xi})^2} \frac{\partial^2 u_\varepsilon}{\partial \eta^2} + \frac{\varepsilon^{1-\alpha}}{1 - \varepsilon^\alpha \bar{\xi}} \frac{\partial u_\varepsilon}{\partial \bar{\xi}} - \varepsilon^{1-2\alpha} \frac{\partial^2 u_\varepsilon}{\partial \bar{\xi}^2} + \varepsilon^{-\alpha} \sin\eta \frac{\partial u_\varepsilon}{\partial \bar{\xi}} - \frac{\cos\eta}{1 - \varepsilon^\alpha \bar{\xi}} \frac{\partial u_\varepsilon}{\partial \eta} = 0.$$

By extracting the leading-order terms, we obtain

$$-\varepsilon^{1-2\alpha} \frac{\partial^2 u_\varepsilon}{\partial \bar{\xi}^2} + \varepsilon^{-\alpha} \sin\eta \frac{\partial u_\varepsilon}{\partial \bar{\xi}} = 0.$$

Setting $\alpha = 1$ yields a boundary layer equation of Prandtl's type,

$$-\frac{\partial^2 \theta^0}{\partial \bar{\xi}^2} + \sin\eta \frac{\partial \theta^0}{\partial \bar{\xi}} = 0, \quad \text{for } 0 < \bar{\xi} < \infty, \quad \pi < \eta < 2\pi,$$

with boundary conditions

$$\theta^0(\eta, \bar{\xi}) = -u^0(\cos\eta, \sin\eta) \quad \text{at } \bar{\xi} = 0,$$

$$\theta^0(\eta, \bar{\xi}) \to 0 \quad \text{as } \bar{\xi} \to \infty.$$

Solving this equation, we obtain the explicit corrector function:

$$\theta^0(\eta, \xi) = -u^0(\cos\eta, \sin\eta) \exp\left(\frac{\sin\eta}{\varepsilon} \xi\right) \chi_{[\pi, 2\pi]}(\eta).$$

This corrector accounts for the boundary layer effects, showing that the thickness of the boundary layer varies with $\eta$ and exhibits degeneracy at $(\pm 1, 0)$. This highlights the necessity of singular perturbation analysis to correctly model such behavior.

## C  CONVERGENCE ANALYSIS OF ENRICHED FEONET

In this section, we will conduct a convergence analysis of the original FEONet to provide theoretical justification for the proposed numerical method. To present the proof clearly, we will restrict our focus to self-adjoint equations with homogeneous Dirichlet boundary conditions:

$$\begin{aligned} -\varepsilon \mathrm{div}(\boldsymbol{a}(\mathbf{x}) \nabla u_\varepsilon) + c(\mathbf{x}) u_\varepsilon &= f(\mathbf{x}) \quad \text{in } D, \\ u_\varepsilon &= 0 \qquad \text{on } \partial D, \end{aligned} \tag{13}$$

where $\boldsymbol{a}(\mathbf{x})$ is a uniformly elliptic coefficient and $c(\mathbf{x}) \geq 0$, which guarantees the well-posedness of the problem. It is noteworthy that the analysis can be easily extended to more general cases (see, e.g., (Hong et al., 2024)).

As described earlier, we let an external forcing term $f$ as the input of neural networks, that is parametrized by $\boldsymbol{\omega}$ in the probability space $(\Omega, \mathcal{F}, \mathbb{P})$. In the convergence analysis, we shall interpret $f(\mathbf{x}; \boldsymbol{\omega})$ as a bivariate function defined on $D \times \Omega$. Moreover, we will assume that

$$f(\mathbf{x}; \boldsymbol{\omega}) \in C(\Omega; L^1(D)) := \left\{ f : \Omega \to L^1(D) : \sup_{\boldsymbol{\omega} \in \Omega} \int_D |f(\mathbf{x}; \boldsymbol{\omega})| \, d\mathbf{x} < \infty \right\}. \tag{14}$$

For each $\boldsymbol{\omega} \in \Omega$, the external force $f(\mathbf{x}; \boldsymbol{\omega})$ is specified, and the corresponding weak solution is denoted by $u_\varepsilon(\mathbf{x}; \boldsymbol{\omega})$, which satisfies the variational formulation:

$$B[u_\varepsilon, v] := \varepsilon \int_D [\boldsymbol{a}(\mathbf{x}) \nabla u_\varepsilon \cdot \nabla v + c(\mathbf{x}) u_\varepsilon v] \, d\mathbf{x} = \int_D f(\mathbf{x}) v \, d\mathbf{x} =: \ell(v) \quad \forall v \in H_0^1(D). \tag{15}$$

For given mesh size $h > 0$, let $\overline{V}_h \subset H_0^1(D)$ be a finite-dimensional space spanned by the basis functions $\{\phi_k\}_{k=0}^{N(h)}$ including the corrector basis function $\phi_0 = \phi_{\mathrm{cor}}$, and $u_{\varepsilon,h}^{\mathrm{en}} \in \overline{V}_h$ be an enriched finite element approximation of $u_\varepsilon$ which satisfies the enriched Galerkin approximation

$$B[u_{\varepsilon,h}^{\mathrm{en}}, v_h] = \ell(v_h) \quad \forall v_h \in \overline{V}_h. \tag{16}$$

We write

$$u_{\varepsilon,h}^{\text{en}}(\mathbf{x}, \boldsymbol{\omega}) = \sum_{k=0}^{N(h)} \alpha_k^*(\boldsymbol{\omega}) \phi_k(\mathbf{x}), \tag{17}$$

where $\alpha^*$ is the finite element coefficients obtained from the linear algebraic system

$$A\alpha^* = F, \tag{18}$$

with

$$A_{ik} = B[\phi_k, \phi_i] \quad \text{and} \quad F_i = \ell(\phi_i). \tag{19}$$

Note that $\alpha^*$ can also be characterized in an alternative way:

$$\alpha^* = \underset{\alpha \in C(\Omega, \mathbb{R}^{N(h)+1})}{\arg\min} \mathcal{L}(\alpha), \tag{20}$$

where $\mathcal{L}$ is the population risk

$$\mathcal{L}(\alpha) = \mathbb{E}_{\boldsymbol{\omega} \sim \mathbb{P}_\Omega} \left[ \sum_{i=0}^{N(h)} |B[\widehat{u}(\boldsymbol{\omega}), \phi_i] - \ell(\phi_i; (\boldsymbol{\omega}))|^2 \right] = \|A\alpha(\boldsymbol{\omega}) - F(\boldsymbol{\omega})\|_{L^2(\Omega)}^2. \tag{21}$$

Next, we define the class of feed-forward neural networks as $\mathcal{N}_n$, where the subscript $n$ denotes the network architecture. We assume that $\mathcal{N}_{n_2}$ is more expressive than $\mathcal{N}_{n_1}$ when $n_1 \leq n_2$. For instance, $n$ could represent the number of layers with bounded width, or the number of neurons when the number of layers is fixed. Neural networks are known to be an appropriate choice for nonlinear approximation, supported by the universal approximation theorem (see, for example, (Cybenko, 1989; Hornik, 1991; Pinkus, 1999; Kidger & Lyons, 2020)). For our analysis in this section, we assume that all neural networks under consideration have a bounded activation function in the final layer (e.g., sigmoid, tanh, etc.), ensuring that the resulting networks are uniformly bounded. Using a straightforward scaling argument, we can show that the universal approximation theorem still applies to this modified class of networks, as discussed in Theorem 2.2 in (Ko et al., 2022).

Now for a neural-network approximation of $\alpha^*$, we mean that $\widehat{\alpha}(n) : \Omega \to \mathbb{R}^{N(h)+1}$, which solves the following minimization problem

$$\widehat{\alpha}(n) = \underset{\alpha \in \mathcal{N}_n}{\arg\min} \mathcal{L}(\alpha), \tag{22}$$

and we write the corresponding solution prediction by

$$\widehat{u}_{\varepsilon,h,n}^{\text{en}}(\mathbf{x}; \boldsymbol{\omega}) = \sum_{k=0}^{N(h)} \widehat{\alpha}(n)_k(\boldsymbol{\omega}) \phi_k(\mathbf{x}). \tag{23}$$

Note here that for the neural network $\alpha \in \mathcal{N}_n$, the input is $\boldsymbol{\omega} \in \Omega$ that specifies the external forcing term $f(\mathbf{x}; \boldsymbol{\omega})$ and the output is the coefficient vector in $\mathbb{R}^{N(h)+1}$.

Finally, we define the solution of the following discrete minimization problem:

$$\widehat{\alpha}(n, M) = \underset{\alpha \in \mathcal{N}_n}{\arg\min} \mathcal{L}^M(\alpha). \tag{24}$$

Here $\mathcal{L}^M$ is the empirical risk, which is the Monte–Carlo integration of the population risk $\mathcal{L}(\alpha)$:

$$\mathcal{L}^M(\alpha) = \frac{|\Omega|}{M} \sum_{m=1}^{M} \sum_{i=0}^{N(h)} |B[\widehat{u}(\boldsymbol{\omega}_m), \phi_i] - \ell(\phi_i; (\boldsymbol{\omega}_m))|^2 = \frac{|\Omega|}{M} \sum_{m=1}^{M} |A\alpha(\boldsymbol{\omega}_m) - F(\boldsymbol{\omega}_m)|^2, \tag{25}$$

where $\{\boldsymbol{\omega}_n\}_{m=1}^M$ is an i.i.d. random variables following $\mathbb{P}_\Omega$. We then write the associated solution as

$$\widehat{u}_{\varepsilon,h,n,M}^{\text{en}}(\mathbf{x}; \boldsymbol{\omega}) = \sum_{k=0}^{N(h)} \widehat{\alpha}(n, M)_k(\boldsymbol{\omega}) \phi_k(\mathbf{x}), \tag{26}$$

which is the actual solution prediction by eFEONet. In the present paper, we assume that we can always find the exact minimizers for the problems equation 22 and equation 24, and the optimization error is ignorable.

To establish suitable theoretical backgrounds for the eFEONet, it is reasonable to prove that the true solution is close enough to the solution prediction computed by the proposed method for various external forces, as the index $n, M \in \mathbb{N}$ goes to infinity. It can be formally written as

$$\|u_\varepsilon - \widehat{u}_{\varepsilon,h,n,M}^{\text{en}}\|_{L^2(\Omega;L^2(D))} \to 0 \quad \text{as } h \to 0 \text{ and } n, M \to \infty. \tag{27}$$

The above total error is divided into three parts:

$$u_\varepsilon - \widehat{u}_{\varepsilon,h,n,M}^{\text{en}} = (u_\varepsilon - u_{\varepsilon,h}^{\text{en}}) + (u_{\varepsilon,h}^{\text{en}} - \widehat{u}_{\varepsilon,h,n}^{\text{en}}) + (\widehat{u}_{\varepsilon,h,n}^{\text{en}} - \widehat{u}_{\varepsilon,h,n,M}^{\text{en}}). \tag{28}$$

The first error arises from the finite element approximation, which we assume to be negligible when $h > 0$ is sufficiently small. In fact, based on the estimate equation 7, we can reduce this error to any desired level by selecting a suitable $h > 0$. Therefore, we assume that $h$ has been chosen so that the finite element approximation error is small enough. The second error, known as the *approximation error*, occurs when we use a class of neural networks to approximate the target (finite element) coefficients. The third error, often referred to as the *generalization error*, measures how well our approximation performs on unseen data. Our focus will be on proving that, with fixed $h > 0$ and $\varepsilon > 0$, as the index $n \in \mathbb{N}$ for neural network architectures becomes larger and the number of input samples $M \in \mathbb{N}$ increases, our approximate solution $\widehat{u}_{\varepsilon,h,n,M}^{\text{en}}$ converges to the finite element solution $u_{\varepsilon,h}^{\text{en}}$ which is assumed to be the true solution here.

## C.1 APPROXIMATION ERROR

First, from equation 21 and equation 25, we observe that the matrix $A$ defined in equation 17 and equation 18 plays a key role in determining the structure of the loss functions. Hence, it would be beneficial for us to analyze these loss functions by understanding more about the matrix. The matrix $A$ is determined by various factors such as the structure of the differential equations, the choice of basis functions, and the boundary conditions. Thus, achieving a characterization of $A$ that is useful for analyzing the loss function and applicable across a wide variety of PDE scenarios is important. The next lemma, quoted from (Ko et al., 2022), addresses this point.

**Lemma C.1.** *Let $A \in \mathbb{R}^{(N(h)+1)\times(N(h)+1)}$ be symmetric and invertible, and we write $\rho_{\min} = \min_i\{|\lambda_i|\}$, $\rho_{\max} = \max_i\{|\lambda_i|\}$ where $\{\lambda_i\}$ is the set of eigenvalues of $A$. Then there holds for any $\mathbf{x} \in \mathbb{R}^{N(h)+1}$ that*

$$\rho_{\min}|\mathbf{x}| \le |A\mathbf{x}| \le \rho_{\max}|\mathbf{x}|. \tag{29}$$

Since the equation 13 is self-adjoint, the associated bilinear form $B[\cdot, \cdot]$ defined in equation 15 is symmetric, which ensures that the matrix $A$ is also symmetric. Additionally, because the coefficient $\boldsymbol{a}(\cdot)$ is uniformly elliptic and $c(\cdot)$ is non-negative, the bilinear form $B[\cdot, \cdot]$ is coercive, meaning that $A$ is positive-definite. As a result, we can apply Lemma C.1 to our enriched finite element matrix $A$.

With this, we are now ready to prove that the approximation error for neural networks converges to zero, as stated in the following theorem.

**Theorem C.2.** *Suppose that the assumption equation 14 holds. Then there holds that*

$$\|\alpha^* - \widehat{\alpha}(n)\|_{L^2(\Omega)} \to 0 \quad \text{as } n \to \infty. \tag{30}$$

*Proof.* Since $A$ is symmetric and positive-definite (and hence invertible), from Proposition C.1, we have that

$$\|\alpha^* - \widehat{\alpha}(n)\|_2^2 \lesssim \|A\alpha^* - A\widehat{\alpha}(n)\|_2^2 \lesssim \|A\alpha^* - F\|_2^2 + \|A\widehat{\alpha}(n) - F\|_2^2 = \mathcal{L}(\widehat{\alpha}(n)) \le \inf_{\alpha \in \mathcal{N}_n} \mathcal{L}(\alpha)$$

$$= \inf_{\alpha \in \mathcal{N}_n} \|A\alpha - F\|_2^2 \lesssim \inf_{\alpha \in \mathcal{N}_n} \left(\|A\alpha - A\alpha^*\|_2^2 + \|A\alpha^* - F\|_2^2\right) = \inf_{\alpha \in \mathcal{N}_n} \|\alpha - \alpha^*\|_2^2.$$

Note that the implicit constants in the above inequalities may depend on $\varepsilon > 0$ and $h > 0$, but are independent of $n \in \mathbb{N}$. As a final step, from the universal approximation property, the last term $\inf_{\alpha \in \mathcal{N}_n} \|\alpha - \alpha^*\|_2^2$ converges to zero as $n \to \infty$. $\qquad\square$

## C.2 GENERALIZATION ERROR

We begin with the definition of *Rademacher complexity*, which measures how the given function class can fit random noise (Gnecco & Sanguineti, 2008; Wainwright, 2019; Bartlett & Mendelson, 2002; Shalev-Shwartz & Ben-David, 2014).

**Definition C.3.** For a family $\{X_i\}_{i=1}^M$ of i.i.d. random variables, the Rademacher complexity of the function class $\mathcal{G}$ is defined by

$$R_M(\mathcal{G}) = \mathbb{E}_{\{X_i,\varepsilon_i\}_{i=1}^M}\left[\sup_{f\in\mathcal{G}}\left|\frac{1}{M}\sum_{i=1}^M \varepsilon_i f(X_i)\right|\right],$$

where $\varepsilon_i$'s are i.i.d. Bernoulli random variables meaning that $\mathbb{P}(\varepsilon_i = 1) = \mathbb{P}(\varepsilon_i = -1) = \frac{1}{2}$ for all $i = 1, \ldots, M$.

Next, we will establish the relationship between the generalization error and the Rademacher complexity for the uniformly bounded function class $\mathcal{G}$. In the following theorem, we assume that the function class is $b$-uniformly bounded, meaning that for any function $f \in \mathcal{G}$, we have $\|f\|_\infty \leq b$.

**Theorem C.4.** [Theorem 4.10 in (Wainwright, 2019)] *Suppose that the family of functions $\mathcal{G}$ is $b$-uniformly bounded. Then for arbitrary small $\delta > 0$, there holds*

$$\sup_{f\in\mathcal{G}}\left|\frac{1}{M}\sum_{i=1}^M f(X_i) - \mathbb{E}[f(X)]\right| \leq 2R_M(\mathcal{G}) + \delta,$$

*with probability at least $1 - \exp(-\frac{M\delta^2}{2b^2})$.*

Next, let us define the following function class:

$$\mathcal{G}_n := \{|A\alpha - F|^2 : \alpha \in \mathcal{N}_n\}, \tag{31}$$

where $A$ and $F$ were defined in equation 19. Then from Lemma C.1, we obtain that

$$\|A\alpha - F\|_{L^\infty(\Omega)} \leq \|A\alpha\|_{L^\infty(\Omega)} + \|F\|_{L^\infty(\Omega)} \lesssim \|\alpha\|_{L^\infty(\Omega)} + \|f\|_{C(\Omega;L^1(D))}.$$

Since the class of neural networks we are considering is uniformly bounded and equation 14 holds, it follows that for any $n \in \mathbb{N}$, the class $\mathcal{G}_n$ is $\tilde{b}$-uniformly bounded for some constant $\tilde{b} > 0$. The following lemma directly follows from Theorem C.4 in our context.

**Lemma C.5.** *Assume that $\{\boldsymbol{\omega}_m\}_{m=1}^M$ is a set of i.i.d. random samples selected from the distribution $\mathbb{P}_\Omega$. Then for any small $\delta > 0$, we have with probability at least $1 - 2\exp(-\frac{M\delta^2}{32\tilde{b}^2})$ that*

$$\sup_{\alpha\in\mathcal{N}_n}\left|\mathcal{L}^M(\alpha) - \mathcal{L}(\alpha)\right| \leq 2R_M(\mathcal{G}_n) + \frac{\delta}{2}. \tag{32}$$

Using Lemma C.5, we now establish the following convergence result for the generalization error. Note here that we assume the Rademacher complexity of $\mathcal{G}_n$ tends to zero as $M \to \infty$, which holds true in many cases (Gnecco & Sanguineti, 2008; Wainwright, 2019; Bartlett & Mendelson, 2002; Shalev-Shwartz & Ben-David, 2014).

**Theorem C.6.** *Assume that equation 14 holds and for any $n \in \mathbb{N}$, $\lim_{M\to\infty} R_M(\mathcal{G}_n) = 0$. Then with probability 1, we have that*

$$\lim_{n\to\infty}\lim_{M\to\infty}\|\widehat{\alpha}(n, M) - \widehat{\alpha}(n)\|_{L^2(\Omega)} = 0.$$

*Proof.* From equation 22 and Proposition C.1, we have

$$\|\widehat{\alpha}(n) - \widehat{\alpha}(n, M)\|_2^2 \lesssim \|A\widehat{\alpha}(n) - A\widehat{\alpha}(n, M)\|_2^2 \lesssim \left(\|A\widehat{\alpha}(n) - F\|_2^2 + \|A\widehat{\alpha}(n, M) - F\|_2^2\right)$$
$$= \mathcal{L}(\widehat{\alpha}(n)) + \mathcal{L}(\widehat{\alpha}(n, M)) \lesssim \mathcal{L}(\widehat{\alpha}(n, M)). \tag{33}$$

We next use Lemma C.5 for $\delta = 2M^{-\frac{1}{2}+\varepsilon}$ with $0 < \varepsilon < \frac{1}{2}$. Then with probability at least $1 - 2\exp(-\frac{M^{2\varepsilon}}{8\tilde{b}^2})$, we obtain that

$$\mathcal{L}(\widehat{\alpha}(n, M)) \leq \mathcal{L}^M(\widehat{\alpha}(n, M)) + 2R_M(\mathcal{G}_n) + M^{-\frac{1}{2}+\varepsilon} \leq \mathcal{L}^M(\widehat{\alpha}(n)) + 2R_M(\mathcal{G}_n) + M^{-\frac{1}{2}+\varepsilon}.$$

By applying Lemma C.5 once more, we have that

$$\mathcal{L}(\widehat{\alpha}(n, M)) \leq \mathcal{L}(\widehat{\alpha}(n)) + 4R_M(\mathcal{G}_n) + 2M^{-\frac{1}{2}+\varepsilon}.$$

With the argument used for the approximation error analysis before, we finally conclude that

$$\lim_{n\to\infty}\lim_{M\to\infty}\|\widehat{\alpha}(n, M) - \widehat{\alpha}(n)\|_2^2 \lesssim \lim_{n\to\infty}\mathcal{L}(\widehat{\alpha}(n)) \lesssim \lim_{n\to\infty}\inf_{\alpha\in\mathcal{N}_n}\|\alpha - \alpha^*\|_2^2 = 0.$$

$\square$

## C.3 MAIN THEORETICAL RESULT ON THE CONVERGENCE OF eFEONet

Combining Theorem C.2 and Theorem C.6, we see that

$$\lim_{n\to\infty} \lim_{M\to\infty} \|\alpha^* - \widehat{\alpha}(n, M)\|_{L^2(\Omega)} = 0. \tag{34}$$

Now we state and prove the main convergence result.

**Theorem C.7** (Convergence of eFEONet). *Assume that equation 14 holds and for any $n \in \mathbb{N}$, $R_M(\widetilde{\mathcal{G}}_n) \to 0$ as $M \to \infty$, where $\widetilde{\mathcal{G}}_n := \{|A\alpha - F|^2 : \alpha \in \mathcal{N}_n\}$. Then for given $\varepsilon > 0$ and $h > 0$, with probability $1$, we have that*

$$\lim_{n\to\infty} \lim_{M\to\infty} \|u_{\varepsilon,h}^{\text{en}} - \widehat{u}_{\varepsilon,h,n,M}^{\text{en}}\|_{L^2(\Omega;L^2(D))} = 0. \tag{35}$$

*Proof.* By Theorem C.2, Theorem C.6, there holds for fixed $\varepsilon > 0$ and $h > 0$ that

$$\|u_{\varepsilon,h}^{\text{en}} - \widehat{u}_{\varepsilon,h,n,M}^{\text{en}}\|_{L^2(\Omega;L^2(D))}^2 = \int_\Omega \int_D \left| \sum_{i=0}^{N(h)} (\alpha_i^* - \widehat{\alpha}(n, M)_i)\phi_i \right|^2 \mathrm{d}\mathbf{x}\,\mathrm{d}\boldsymbol{\omega}$$

$$\leq \int_\Omega \int_D \sum_{i,j=0}^{N(h)} |\alpha_i^* - \widehat{\alpha}(n, M)_i|^2 |\phi_i|^2 \,\mathrm{d}\mathbf{x}\,\mathrm{d}\boldsymbol{\omega}$$

$$+ \int_\Omega \int_D \sum_{i,j=0}^{N(h)} |\alpha_j^* - \widehat{\alpha}(n, M)_j|^2 |\phi_j|^2 \,\mathrm{d}\mathbf{x}\,\mathrm{d}\boldsymbol{\omega}$$

$$\leq \int_\Omega \int_D 2N(h) \sum_{k=0}^{N(h)} |\alpha_k^* - \widehat{\alpha}(n, M)_k|^2 |\phi_k|^2 \,\mathrm{d}\mathbf{x}\,\mathrm{d}\boldsymbol{\omega}$$

$$\lesssim \|\alpha^* - \widehat{\alpha}(n, M)\|_{L^2(\Omega)}^2,$$

where all the implicit constants above are independent of $n$, $M \in \mathbb{N}$. Taking $n, M \to \infty$, we complete the proof. $\square$

*Remark* C.8. It is noteworthy that the convergence in Theorem C.7 is not uniform with respect to $h \to 0$. Indeed, this issue aligns precisely with the main theme of reference (Hong et al., 2024), where the authors rigorously demonstrated that both the approximation error and generalization error depend on the condition number $\kappa(A)$ of the finite element matrix $A$. Typically, $\kappa(A) \sim h^{-2}$, meaning that as $h$ becomes smaller, both the approximation and generalization errors increase due to this adverse dependence. To summarize, the total error can be characterized as:

$$(\text{Total Error}) \lesssim h^\alpha + \frac{h^{-\beta}}{\sqrt{n}} + \frac{h^{-\gamma}}{\sqrt{M}},$$

for some positive constants $\alpha$, $\beta$ and $\gamma$. In the regime where $h$ is not too small, the first term dominates, and the total error decreases with decreasing $h$. However, beyond a certain threshold, the last two terms begin to dominate, causing the total error to increase. One can mitigate this phenomenon by utilizing several strategies. For instance, employing higher-order FEM increases $\alpha$, thereby reducing the approximation error. Alternatively, increasing $n$ and $M$ reduces the generalization error. Most importantly, one can use the preconditioning techniques to reduce $\kappa(A)$, thereby significantly diminishing the last two components. While this analysis was originally developed in the context of FEONet, it applies directly to eFEONet as well, since in eFEONet we solve equations with a fixed small $\varepsilon$. More precisely, the only part of the analysis in (Hong et al., 2024) where $\varepsilon$ could potentially affect the results is in the condition number estimates (Eq. (2.11) and (2.12) on page 6 of (Hong et al., 2024)). If we explicitly characterize the dependency on $\varepsilon$ in these equations, then we can likewise make the $\varepsilon$-dependence explicit in the final error estimate (Theorem 4.10 on page 18). In doing so, we can obtain a complete error analysis for eFEONet that incorporates both singular perturbation asymptotic analysis and the general framework from (Hong et al., 2024), which will be addressed in the forthcoming paper.

# D    EXPERIMENT DETAILS

## D.1    RANDOM GENERATION OF INPUT FUNCTIONS FOR EXTERNAL FORCE

In order to train neural networks, we need to generate random external forcing functions. Inspired by Bar-Sinai et al. (2019), we created a random signal $f(\mathbf{x}; \boldsymbol{\omega})$ as a linear combination of sine functions and cosine functions. More precisely, we set

$$f(x) = m_0 \sin(n_0 x) + m_1 \cos(n_1 x) \tag{36}$$

for 1D cases and

$$f(x, y) = m_0 \sin(n_0 x + n_1 y) + m_1 \cos(n_2 x + n_3 y) \tag{37}$$

for 2D cases where $m_i$ for $i = 1, 2$ and $n_j$ for $j = 0, 1, 2, 3$ are drawn independently from the uniform distributions. It is worth noting that even when considering different random input functions, such as those generated by Gaussian random fields, we consistently observe similar results. This robustness indicates the reliability and stability of the eFEONet approach across various input scenarios.

For Section 4.4, we randomly sample the external force as $f(x, y) = (1 - x^2)^2 \big[ m_0 \sin(n_0 x) + m_1 \cos(n_1 y) \big]$.

## D.2    EXPERIMENT SETTINGS

In this section, we outline the experimental setup. For the problems under consideration, we used the neural network, which consists of 6 convolutional layers with swish activation, followed by a fully connected layer flattening the output. For the 1D problems, we used Conv1D, while Conv2D was used for 2D problems. The eFEONet was trained with the LBFGS optimizer along with the following hyperparameters.

- Maximal number of iterations per optimization step: 100;
- Learning rate : 0.1
- Update history size: 100.

We used the Intel Xeon Gold 6226R processor and NVIDIA RTX A6000 48GB GPU.

For the 1D problems, the training dataset for FNO and ComFNO includes $900 \times 201$ tuples $(f, u)$, while the 2D scenarios encompass $900 \times 51 \times 51$ tuples $(f, u)$ as described in the paper (Li et al., 2024). In all conducted experiments, we utilized the mean-square loss functions. For FNO and ComFNO, we used the Adam optimizer for all minimization problems, accompanied by the consistent utilization of the GELU activation function. Further details concerning the remaining parameters for our result can be found in Table 4 and Table 5.

| Experiment/FNO | depth | LR | epoch | batch size |
|---|---|---|---|---|
| 1D (no turning point) | 4 | 0.001 | 500 | 50 |
| 1D (turning point) | 6 | 0.001 | 500 | 50 |
| 2D | 5 | 0.001 | 1000 | 50 |

Table 4: Experimental parameters for FNO investigations. The term "depth" denotes the quantity of Fourier layers implemented within the architecture. "LR" designates the learning rate employed, while "epoch" signifies the count of training iterations performed.

| Experiment/ComFNO | blockNum | LR | epoch | batch size |
|---|---|---|---|---|
| 1D (no turning point) | 1 | 0.001 | 500 | 30 |
| 1D (turning point) | 2 | 0.001 | 500 | 30 |
| 2D | 2 | 0.001 | 1000 | 20 |

Table 5: Experimental parameters for ComFNO investigations. The term "blockNum" denotes the quantity of layer blocks implemented within the architecture. "LR" designates the learning rate employed, while "epoch" signifies the count of training iterations performed.

### D.3 RUNTIME AND EFFICIENCY ANALYSIS

We report the training and inference costs for the 2D convection–diffusion problem, averaging results over 10 independent runs with different random seeds.

- **FNO**:
    - Inference time per sample: 3.951 ms
- **ComFNO**:
    - Inference time per sample: 4.885 ms
- **Our method (eFEONet)**:
    - Inference time per sample (ms): Mean 3.81, Std 3.98, 95% CI [0.961, 6.657]
    - Relative $L^2$ error: Mean $2.26 \times 10^{-3}$, Std $2.92 \times 10^{-4}$, 95% CI $[1.69 \times 10^{-3}, 2.83 \times 10^{-3}]$

## E  FURTHER EXPERIMENTS

### E.1  ORDINARY DIFFERENTIAL EQUATIONS WITH BOUNDARY LAYER

We consider the 1D singularly perturbed differential equation given by:
$$-\varepsilon u_\varepsilon'' + (x+1)u_\varepsilon' = f(x), \quad x \in (0,1),$$
$$u_\varepsilon(0) = u_\varepsilon(1) = 0. \tag{38}$$

The exact solution $u_\varepsilon$ for equation 38 is approximately given by:
$$u(x) \approx u_0(x) - u_0(1)\exp\left(-2\frac{1-x}{\varepsilon}\right), \tag{39}$$

where $u_0(x)$ is the reduced solution, and $\exp(-2\frac{1-x}{\varepsilon})$ represents the corrector function. Figure 8 illustrates the corrector function for the equation equation 38 as $\varepsilon$ varies from $10^{-1}$ to $10^{-4}$.

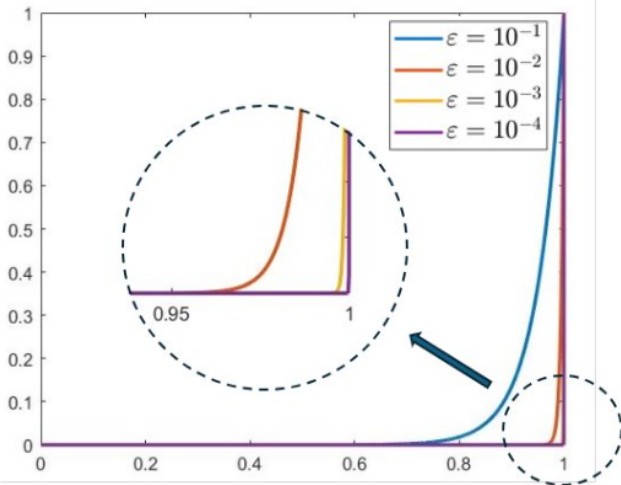

Figure 8: Corrector functions for the equation equation 9 with varying values of $\varepsilon$ from $10^{-1}$ to $10^{-4}$.

As depicted in Figure 8, the solution displays an exponential boundary layer at $x = 1$. An example of the solution to equation 38 is depicted in Figure 10. We evaluate different input-output resolutions, specifically 50 and 200, comparing our predictions with the ground truth obtained from the upwind scheme on the Shishkin mesh. The eFEONet predictions for different resolutions are illustrated in Figure 10. Notably, the results remain consistent even at lower resolutions. The performance of our method for $\varepsilon = 10^{-5}$ with 100 test function $f$ samples is shown in Figure 11. The figure on the left shows the input function $f$, the middle figure shows the ground truth corresponding to the 100 test $f$ samples, and the figure on the right shows the residuals produced by our method for these 100 $f$ samples.

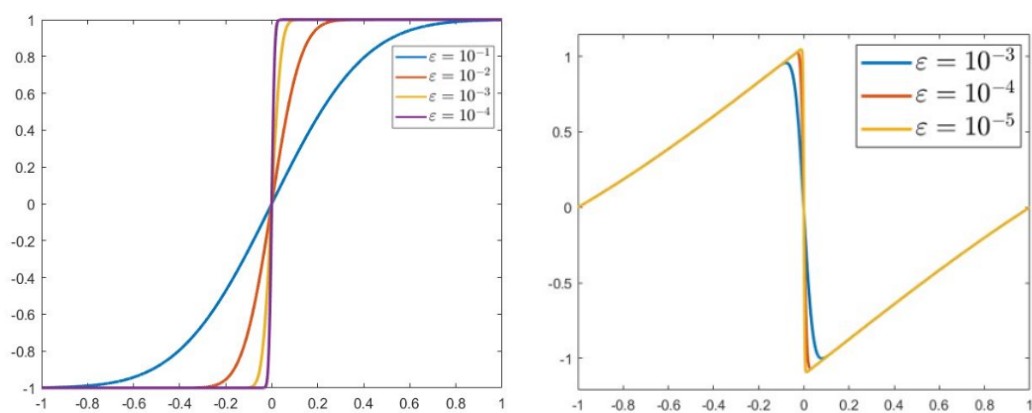

Figure 9: Corrector function for equation 10 with $\varepsilon$ varying from $10^{-1}$ to $10^{-4}$ (left) and the corresponding interior layer structures with $\varepsilon$ changing from $10^{-3}$ to $10^{-5}$ (right).

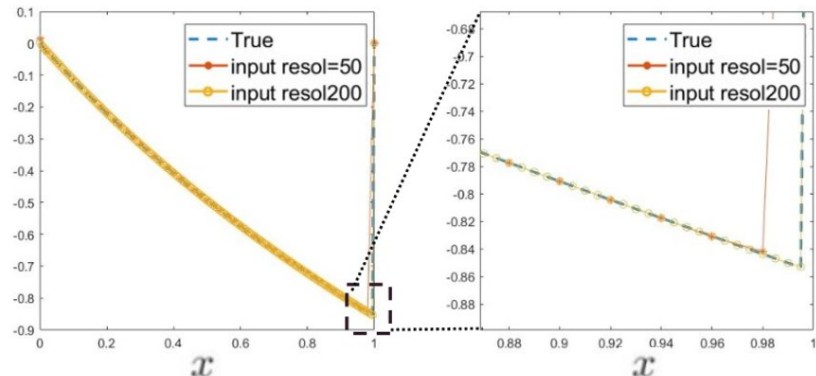

Figure 10: Comparison of the predicted solution $\widehat{u}_\varepsilon$ for each input-output resolution $= 50, 200$ and true solution $u_\varepsilon$

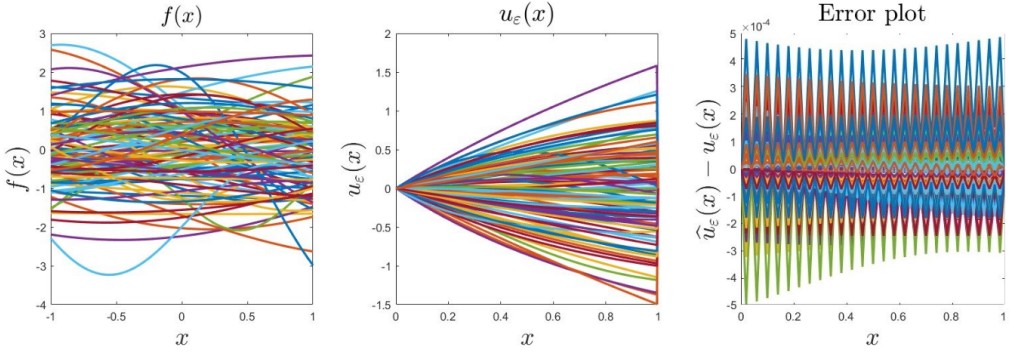

Figure 11: Visualization of 100 input functions $f$ (left), corresponding reference solutions (middle), and error plots (right) for Section 4.1 with $\varepsilon = 10^{-5}$, input-output resolution $= 51$.

### E.2 ORDINARY DIFFERENTIAL EQUATIONS WITH INTERIOR LAYER

When the coefficient of $u'$ vanishes at certain points, we refer to this as a turning point problem. We now consider an example with a single turning point at $x = 0$ over the interval $[-1, 1]$:

$$-\varepsilon u''_\varepsilon - x u'_\varepsilon = f(x), \quad x \in (-1, 1),$$
$$u_\varepsilon(-1) = u_\varepsilon(1) = 0. \quad (40)$$

The corrector function is shown in Figure 9. As depicted in Figure 9, the interior layer structure near $x = 0$ becomes steeper as $\varepsilon$ decreases. Figure 12 presents an example of turning point problems for input-output resolutions of 25, 50, 100, and 200. The results are consistent with the previous case. The performance of our method for $\varepsilon = 10^{-5}$ with 100 test function $f$ samples is shown in Figure 13. The figure on the left shows the input function $f$, the middle figure shows the ground truth corresponding to the 100 test $f$ samples, and the figure on the right shows the residuals produced by our method for these 100 $f$ samples.

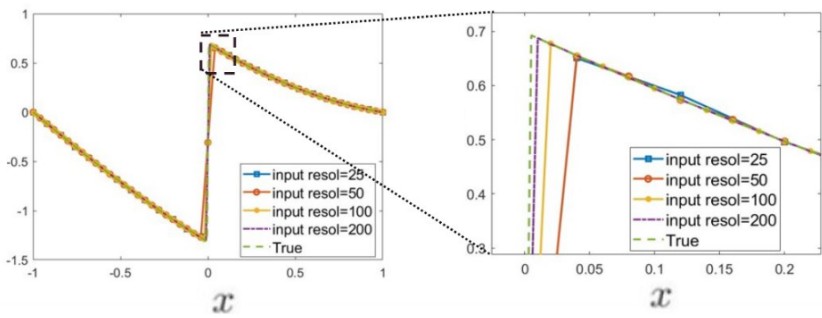

Figure 12: Comparison of the predicted solution $\widehat{u}_\varepsilon$ for each input-output resolution $= 25, 50, 100, 200$ and true solution $u_\varepsilon$

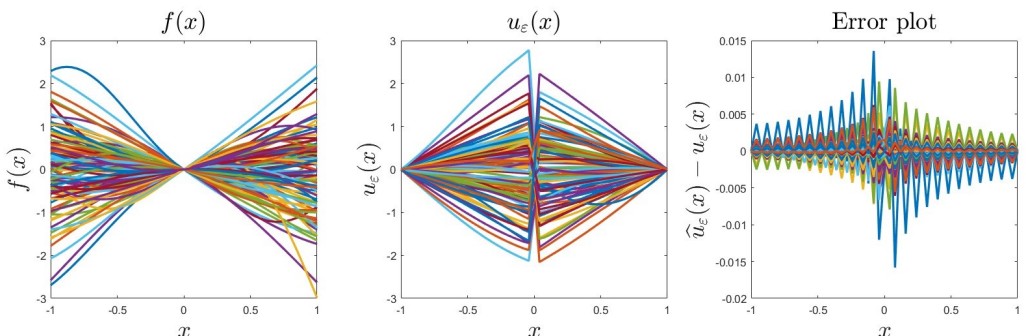

Figure 13: Visualization of 100 input functions $f$ (left), corresponding reference solutions (middle), and error plots (right) for Section 4.2 with $\varepsilon = 10^{-5}$, input-output resolution $= 51$.

### E.3 PARTIAL DIFFERENTIAL EQUATIONS ON SQUARE

We examine a boundary value problem for an elliptic PDE over the spatial domain $D = [0, 1]^2$:

$$-\varepsilon \Delta u_\varepsilon - (1, 1) \cdot \nabla u_\varepsilon = f(x, y) \quad \text{in } D,$$
$$u_\varepsilon(x, y) = 0 \quad \text{on } \partial D, \quad (41)$$

For this PDE problem, the asymptotic expansion of $u(x, y)$ is formulated as:

$$u(x, y) = u_0(x, y) - u_0(0, y)e^{-x/\varepsilon} - u_0(x, 0)e^{-y/\varepsilon} + u_0(0, 0)e^{(-x-y)/\varepsilon}.$$

The solution exhibits a boundary layer along $x = 0$ and $y = 0$, with a corner layer forming at $(0, 0)$, as illustrated in Figure 14. To solve the equation equation 41, we employ both FNO, ComFNO, and eFEONet. Residuals for 100 randomly chosen $f$ sample with $\varepsilon = 10^{-4}$ are presented in Figure 15.

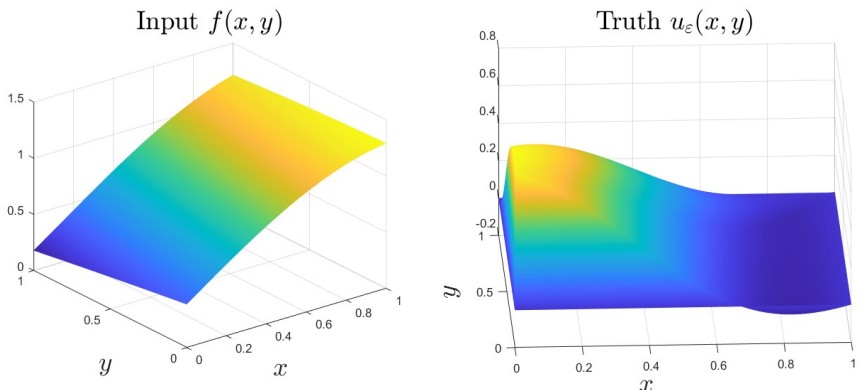

Figure 14: Solution profiles for the PDE problem on a square domain.

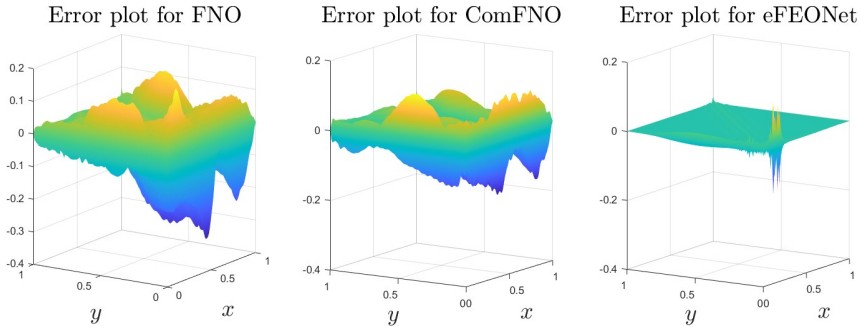

Figure 15: Visualization of error plots from FNO for 100 test function(left), ComFNO(middle) and eFEONet(right) for Section 4.3 with $\varepsilon = 10^{-4}$, input-output resolution $= 51$.

### E.4 PARTIAL DIFFERENTIAL EQUATIONS ON CIRCLE

Finally, we investigate singularly perturbed differential equations in various geometries. As a first case, we consider the equation over the unit circle $D$, given by

$$-\varepsilon\Delta u_\varepsilon - (u_\varepsilon)_y = f(x,y) \quad \text{in } D,$$
$$u_\varepsilon = 0 \quad \text{on } \partial D,$$

where $0 < \varepsilon \ll 1$, $D$ represents the unit disk centered at $(0,0)$, and $f(x,y)$ is a smooth function defined over $D$. Figures 16 and 17 illustrate the solution profiles for different values of $\varepsilon$. Furthermore, our approach can be extended to more complex geometries, as demonstrated in Figure 18.

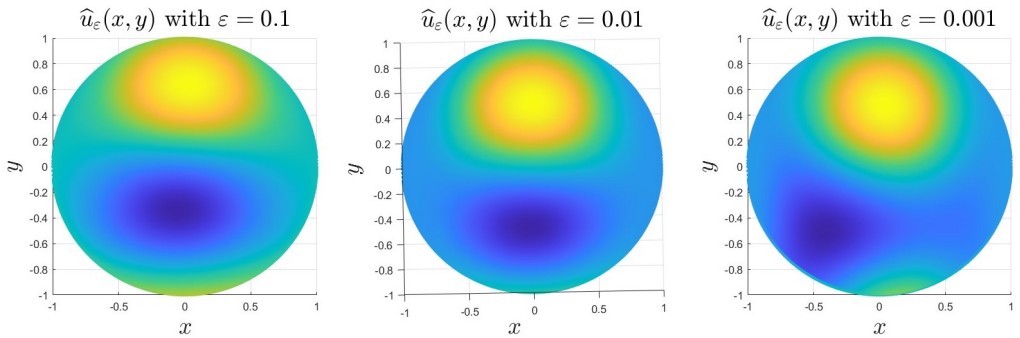

Figure 16: Solution profiles for a circular domain with varying $\varepsilon > 0$ when the source term is given by $f(x,y) = (1-x^2)^2(-0.194\sin(-1.696x) - 1.12\cos(4.052y))$

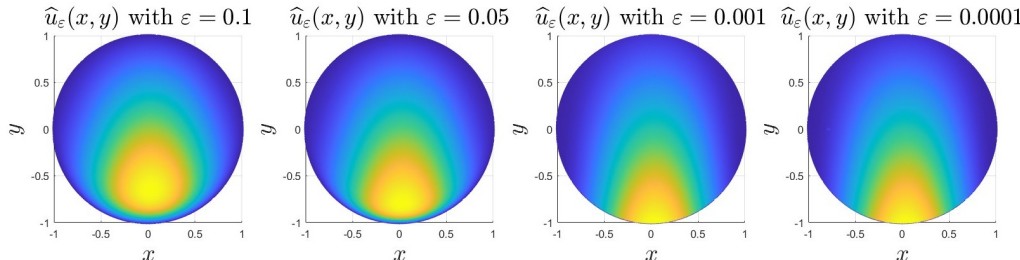

Figure 17: Solution profiles for a circular domain with different $\varepsilon > 0$, where the source term is given by $f(x, y) = (1 - x^2)^2(0.49\sin(1.03x) + 0.727\cos(-0.303y))$

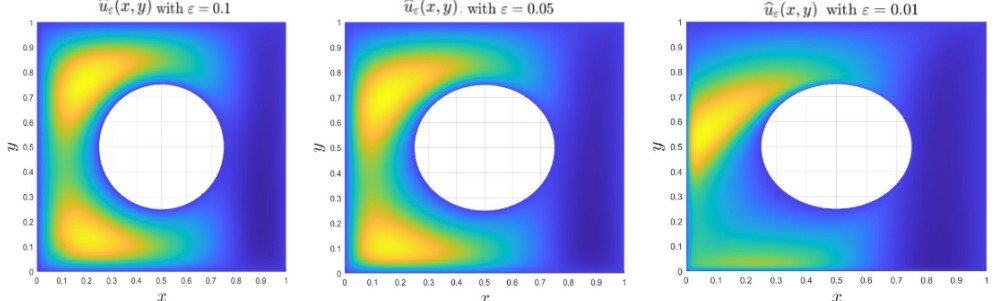

Figure 18: Results for square with a hole with different $\varepsilon > 0$.

### E.5    Generalizability of Corrector Functions

We propose the eFEONet for solving singularly perturbed PDEs without requiring data. Our approach uses a corrector function derived from asymptotic analysis to obtain an accurate solution. Additionally, we present a data-driven approach that simultaneously learns the corrector function and the solution coefficients. We conducted two experiments to validate our methods. Our method integrates a specialized corrector function, $\phi_{\text{cor}}$, into the standard Finite Element Method (FEM) framework. The final solution is represented as a linear combination of basis functions and the corrector function.

- **Experiment 1: Approximating corrector function.** In the first experiment, we attempted to approximate the solution function of a 1D boundary layer problem using a corrector function:

$$\phi_{\text{cor}}(x) := e^{-(1+x)/\varepsilon} - (1 - (1 - e^{-2/\varepsilon})(x+1)/2)$$

This function was approximated by a linear combination of basis functions, $\sum_{i=1}^{N} c_i^c \phi_i$. The goal was to use this approximation to generate the weak formulation.
The approximation error, measured by the $l^\infty$ norm, was relatively small:

$$\|\phi_{\text{cor}} - \sum_{i=1}^{N} c_i^c \phi_i\|_{l^\infty} \approx 3 \times 10^{-3}$$

However, this approach failed because the derivative of the corrector function, $\phi'_{\text{cor}}$, scales with $1/\varepsilon$, leading to large errors in the weak formulation due to the approximation's inability to accurately capture this behavior.

- **Experiment 2: Simultaneous prediction of corrector function and coefficients.** In the second experiment, we used a data-driven approach to simultaneously predict the coefficients and the corrector function. The coefficients were predicted as $c_i + c_i^c \exp(\xi_\theta)$, where the network outputted $c_i$, $c_i^c$, and $\xi_\theta$. The results, summarized in Table 6, demonstrate a significant improvement in accuracy.

Table 6: Comparison of Solution Accuracy

| Method | Error |
|---|---|
| Learnable corrector (data-driven) | $3.2 \times 10^{-3}$ |
| eFEONet | $1 \times 10^{-5}$ |

The results of the second experiment clearly indicate that our data-driven approach, which simultaneously predicts the corrector function and coefficients, is highly effective. It successfully addresses the limitations of simply approximating a theoretically derived corrector function. The significant reduction in error from $3.2 \times 10^{-3}$ to $1 \times 10^{-5}$ demonstrates the superiority of our proposed method.

### E.6    Comparison with the original FEONet

The original FEONet does not incorporate a corrector function, and therefore, its performance is comparable to the standard FEM without any enhancement. In contrast, our proposed eFEONet achieves significantly lower errors in both boundary and interior layer regions, as shown in Table 7.

| Model | Boundary Layer | Interior Layer |
|---|---|---|
| FEONet | 3.04 | 0.0222 |
| Ours (eFEONet) | 7.0e-05 | 6.6e-04 |

Table 7: Comparison of FEONet and eFEONet for $\varepsilon = 10^{-5}$. Errors are reported for the boundary layer and interior layer regions.

## F    Broader Impact Discussion

Our proposed model, eFEONet, is designed to solve partial differential equations (PDEs) efficiently, which has significant potential for positive societal impacts. By providing rapid and accurate computational predictions, our approach can accelerate scientific discovery and engineering advancements

in various fields such as climate modeling, fluid dynamics, and structural engineering. Improved computational efficiency can also contribute to reduced energy consumption and enhanced sustainability in high-performance computing contexts.

However, we recognize potential negative societal impacts arising from inappropriate reliance on model predictions. Specifically, inaccurate or overly confident reliance on model outputs without sufficient validation could lead to erroneous conclusions or misguided decision-making in safety-critical applications. To mitigate these risks, we strongly advocate for rigorous verification and validation processes, transparency in model limitations, and cautious interpretation of the results when deploying computational models in practical scenarios.

