# OpenReview forum: "Data-free Asymptotics-Informed Operator Networks for Singularly Perturbed PDEs"
_ICLR.cc/2026/Conference — ICLR 2026 Conference Withdrawn Submission_

### Official Review · Reviewer_Z7k5 · 2025-10-21

**Soundness:** 3
**Presentation:** 4
**Contribution:** 3
**Rating:** 8
**Confidence:** 5

**Summary:**

This paper introduced a kind of neural operator called eFEONet, to solve singularly perturbed differential equations. Instead of learning the force function to solution function directly, they learn the coefficients of the manually designed basis functions. The basis functions consists of corrector functions getting from asymptotic analysis, making it suitable for singularly perturbed problems. eFEONet can achieve superior accuracy in a unsupervised manner, compared to the supervised FNO and ComFNO. A qualitative convergence analysis is given, and four examples are tested to validate the method.

**Strengths:**

1.I appreciate the idea of integration of existing analytical methods into an operator learning framework to solve harder problems, instead of increasing data and computing power.

2.The method overcomes the data requirement limitation by being highly data-efficient, requiring minimal training data, or even operating without any training dataset.

3.The numerical results show that eFEONet achieves error reductions of two orders of magnitude compared to existing FNO and ComFNO for problems with boundary and interior layers in both 1D and 2D, even in unsupervised manner.

**Weaknesses:**

1.The convergence Theorem 3.3 is qualitative and weak, does not align with the high accuracy in numerical experiments.

2.The eFEONet is not well structured in section 3.2. A detailed algorithm should be given instead of a schematic picture.

3.The experiments seems weak since the four singular perturbed problems can be asymptotically analyzed. The corrector functions are easier for these four problem, the extension to other complex singular perturbed problems should be addressed, i.e. the boundary of the method should be clarified.

**Questions:**

1.Can the convergence result be quantified? How Does the upper bound depend on epsilon, h, n and M? The numerical results seems it is asymptotic convergent with epsilon, can it be theoretically proved? For regular boundary problems, there are works [a] to prove the uniform convergence with epsilon by simply using Shishkin mesh as collocation points in the vanilla DeepONet architecture.
[a] Approximation and generalization of DeepONets for learning operators arising from a class of singularly perturbed problems.

2.The loss (6) is to minimizing the norm of linear algebraic equation’s residual, so why not solve the linear algebraic equation ((18) in the appendix) directly to get the coefficients? It is more accurate than minimize the residual, and solving linear algebraic equation is also faster, then we do not need eFEONet anymore? The advantage compared to this traditional numerical scheme is not well addressed.

---

### Official Review · Reviewer_MSAk · 2025-10-28

**Soundness:** 3
**Presentation:** 3
**Contribution:** 2
**Rating:** 4
**Confidence:** 4

**Summary:**

The paper considers using machine learning methods to solve PDEs, in particular singularly perturbed differential equations, by combining FEM with PINNs. The work extends FEMNet to incorporate corrector basis functions, to better learn asymptotic behavior of solutions. The authors provide theoretical analysis for convergence, and demonstrated in experiments that the method improves prediction error for four types of singularly perturbed differential equations.

**Strengths:**

1. The paper is clearly written and easy to follow.
2. The idea of enriching basis functions with correctors seems to improve PINNs with prior basis for singularly perturbed differential equations.
3. The theoretical analysis seems sound.

**Weaknesses:**

1. The method relies on a previously specified corrector function or corrector function class (as discussed in Appendix E.5). These corrector functions seem to make the key difference with FEMNet and are the key enabler for capturing the fine details in the PDEs considered. However, how to identify such corrector function class is not discussed in the paper, and I believe it is a rather challenging task and the same corrector function might not generalize well across different PDE instances, which greatly limit the applicability of the method.
2. The comparison with baselines might not be fair under the current setting. For the original basis function, the paper proposes to use Galerkin approximation to find such basis. It is well known that Galerkin methods require many pre-generated solution data points to find good basis functions, and it seems that in Appendix D.1 the varying forces are used for that besides training the NN. If this is the case, the proposed method seems to have further limited applicability compared with methods such as FNO, where such a procedure is not required.
3. The experiments lack closely related baseline methods and benchmark problems. Since the proposed method is a natural extension of FEMNet, comparison with this method is needed. Furthermore, if possible I would love to see how the proposed method performs compared with other state-of-the-art methods such as [1] and baselines therein. In addition, there is a lack of experiments on benchmark PDE problems such as ones in [2]. The paper misses the part on how the method performs on not singularly perturbed differential equations to examine its general applicability.

[1] Wu, Haixu, et al. "Transolver: A fast transformer solver for pdes on general geometries." arXiv preprint arXiv:2402.02366 (2024).

[2] Takamoto, Makoto, et al. "Pdebench: An extensive benchmark for scientific machine learning." Advances in Neural Information Processing Systems 35 (2022): 1596-1611.

**Questions:**

1. How fast is the ground truth solver on the problems considered? Is the proposed method faster than that?
2. What are the exact inputs for each method in the experiment? Do the methods predict PDE solutions for fixed parameters or varying parameters for a given model? If the latter, how is the generalization capability of each method?
3. Are there any quantitative results for section 4.4?

---

### Official Review · Reviewer_Bbns · 2025-10-29

**Soundness:** 3
**Presentation:** 2
**Contribution:** 3
**Rating:** 4
**Confidence:** 4

**Summary:**

In this paper, eFEONet is proposed to handle singularity perturbed PDE, which is commonly known to be difficult to solve because of the sharp changes. The proposed method utilizes singular perturbation theory by integrating special basis function into the FEONet framework to better capture the asymptotic behaviors. Experiments on convection-diffusion PDEs in 1D and 2D show the proposed method can achieve error reduction up to 2 orders of magnitude compared to FNO, ComFNO, and DeepONet, while also more data-efficient. A convergence analysis is also provided.

**Strengths:**

- Singularly perturbed PDEs are important in many scientific and engineering domains. This method shows good accuracy compared to both neural-network based and standard FEM methods.
- A convergence analysis is provided for theoretical guarantees.

**Weaknesses:**

- If this work is based on FEONet, then the contribution above FEONet should be better clarified, and FEONet should also be compared in the experiments as a baseline.
- Most experimental settings are relatively simple, i.e., in low dimension, small range, with ground truths of simple shapes.
- The neural network structure is trivial from the perspective of a deep learning society.

**Questions:**

It will be good to see the training and inference computational cost, for both the proposed method and baselines. In Section D.3, only the inference time is given.

---

### Official Review · Reviewer_SkGd · 2025-10-29

**Soundness:** 2
**Presentation:** 2
**Contribution:** 1
**Rating:** 2
**Confidence:** 5

**Summary:**

How to solve singularly perturbed PDEs by using neural network methods is an interesting research topic. However, the proposed method requires to know the exact asymptotic behavior of the solution. However, obtaining the exact asymptotic behavior of the solution is the most challenging part for solving a singularly perturbed PDE. If it is known, the remaining part is relatively simple, either numerical methods or neural networks work well. Besides, the proposed neural network method is not well presented.

**Strengths:**

Although several methods have been proposed, how to solve singularly perturbed PDEs by using neural network methods is still an interesting research topic.

**Weaknesses:**

(i) The proposed method requires to know the exact asymptotic behavior of the solution. However, obtaining the exact asymptotic behavior of the solution is the most challenging part for solving a singularly perturbed PDE. If it is known, the remaining part is relatively simple, either numerical methods or neural networks work well. (ii)  It seems that the authors do not know the difference between operator learning and learning the solution of a single PDE initial/boundary value problem. (iii) Although presented in Figure 3, it is not mentioned in the text where the neural network is used and what are the randomly drawn parameters $\omega_i\in\Omega$ and how they are related to the given data.

**Questions:**

Please see the weaknesses part.

---

### Note · Authors · 2026-01-21

I have read and agree with the venue's withdrawal policy on behalf of myself and my co-authors.